# SPL: Orchestrating Workflows with Declarative Deterministic–Probabilistic Composition

## Abstract

Agentic workflows increasingly need to compose probabilistic LLM inference with deterministic, machine-checkable computation: plan a derivation, then verify it in a symbolic kernel; draft code, then test it against a specification. Declarative LLM-orchestration systems (LMQL, DSPy, and other recent designs) address the first need; symbolic tool-calling frameworks address the second; composing both today still requires hand-written glue code that is neither declarative nor portable. We present SPL (Structured Prompt Language), a declarative language that gives both computation modes - probabilistic (`GENERATE/EVALUATE/WHILE`) and deterministic (`SOLVE/ASSERT`) - first-class status in a single formal grammar sharing one variable namespace, so an LLM's output flows directly into a symbolic kernel check and back without marshaling between two separately-orchestrated systems. A `.spl` specification only declares *what* workflow to run, not *which* model or verifier runs it - so the same script runs unchanged across various language models / backends / runtimes.

We validate SPL with an 80-recipe cookbook - including compilation to four external orchestration targets - and a controlled 1,200-run experiment on symbolic mathematics. The solver arm's kernel-execution pass rate spans 32–93% across ten models; independently reverifying each passing result by substituting it back into the original problem's defining relation confirms 86.7% of these as semantically correct, isolating how much of the raw pass rate reflects execution success alone rather than a solved problem. The LLM-only baseline measures output production, not correctness, and is not directly comparable to this verified rate. SPL is released under Apache 2 license.

## 1 Introduction

### 1.1 The Problem: Fragmentation in LLM Programming

Building LLM-powered workflow systems today demands a daunting intersection of skills. A practitioner must master prompt engineering for effective LLM interaction, Python programming for orchestration logic, API wrangling across disparate model providers, and manual state management for multi-step reasoning. Each popular framework — LangGraph, AutoGen [1], CrewAI [2] — introduces its own abstractions, API surfaces, and execution models, creating a fragmented ecosystem where knowledge transfers poorly and vendor lock-in is the norm.

This situation echoes a pattern the data industry has seen before. In the early days of relational databases, programmers wrote imperative C and COBOL code to traverse data structures, manage cursors, and handle errors — all in application-specific ways. The introduction of SQL in the 1970s transformed this landscape by providing a declarative specification layer: users described *what* data they wanted, not *how* to retrieve it. This separation enabled decades of optimizer innovation underneath a stable surface language. The current

---

[0]Keywords: declarative language, LLM orchestration, deterministic-probabilistic composition, workflow specification, symbolic verification, structured prompt language

state of LLM programming is analogous to the pre-SQL era: powerful capabilities buried under layers of imperative glue code.

The barrier to entry compounds the problem. To build even a simple self-refining workflow — one that drafts, critiques, and revises its own output — a developer must write 84–147 executable lines of Python across multiple framework abstractions.[1] This excludes the vast population of domain experts, analysts, and researchers who understand their workflows but lack the software engineering skills to express them in imperative orchestration code.

## 1.2 The Two-Mode Gap

All existing frameworks — declarative and imperative alike — operate in a single computation mode. LLM-centric systems (LangGraph, DSPy, AutoGen) produce approximate, non-reproducible outputs shaped by the model's training distribution. Symbolic tools (SymPy, SageMath, Lean) produce exact, reproducible, machine-verifiable results. Neither world speaks the other's language within a unified programming model. We call this the **two-mode gap**.

Real agentic tasks routinely require both. A homework assistant must plan a solution in natural language (probabilistic), execute each algebraic step exactly (deterministic), verify the result (deterministic), then explain it to the student (probabilistic). The coupling between modes may be *loose* — distinct phases passing data once — or *tight*: an `ASSERT` failure immediately triggers a corrective `GENERATE`, which revises the plan and re-enters the symbolic engine. Both patterns arise in practice; both require application-specific glue code to coordinate state transfer and error propagation at every mode crossing.

The gap is an expressiveness ceiling, not an engineering inconvenience. No existing workflow language lets the programmer declare "compute this step exactly" and "generate this step approximately" in the same specification. SPL closes it through *declarative composition*: the programmer specifies what each step computes and which mode it inhabits; the runtime resolves the boundary, manages state transfer, and enforces verification gates — without glue code.

The dual-process theory of human cognition [21] offers a useful analogy — but requires a critical correction. Kahneman's framing names the two systems *fast* and *slow*, conflating epistemic mode with cognitive speed. That conflation misleads in a computational setting: discovering quantum mechanics was intuitive, exploratory, and extraordinarily slow; once Schrödinger's equation was established, *solving* it became deterministic and mechanical — slow by hand, fast by computer. Speed tracks the implementation, not the mode.

The relevant distinction is epistemic, not temporal. System 1 is *probabilistic and intuitive*: it pattern-matches, generates hypotheses, and produces approximate outputs shaped by prior experience — regardless of wall-clock time. System 2 is *deterministic and mechanical*: it applies explicit rules to produce exact, reproducible results — regardless of whether those rules execute in milliseconds or minutes. In SPL's mapping: `GENERATE` and `EVALUATE` are System 1 primitives (LLM inference, approximate, quality-assessed); `SOLVE` and `ASSERT` are System 2 primitives (symbolic kernel, exact, machine-verifiable). A SymPy kernel may compute a derivative faster than the LLM does; a Lean 4 proof may take longer. Neither fact changes which mode each step inhabits. The organizing principle is mode, not speed.

## 1.3 Our Contributions

We present SPL, a declarative language that composes both computation modes in a single coherent specification. The primary contributions of this paper are:

**C1. Declarative two-mode composition.** SPL provides `SOLVE` and `ASSERT` — primitives that route computation to a live Python kernel (IPython) rather than an LLM — alongside `GENERATE` and `EVALUATE`

---

[1]Measured on the self-refine recipe (`cookbook/05_self_refine/`), which ships hand-built equivalents for four frameworks: AutoGen 84 executable LOC (118 lines total), CrewAI 90 (125 total), LangGraph 107 (158 total), PocketFlow 147 (225 total). Executable lines exclude blank lines and single-line comments; total lines include both. The equivalent `self_refine.spl` is 95 lines total, of which approximately 35 are executable SPL control-flow instructions; the remainder are three LLM prompt templates embedded as string literals (the `AS $$...$$` blocks) and blank lines.

for the probabilistic mode. A single declarative specification freely composes both modes; variable bindings flow between them through a shared `@variable` namespace without programmer-managed marshaling. The mode boundary is visible in the source and managed entirely by the runtime. This declarative structure is not a virtue in itself: because the specification names only *what* each step computes, not *which* model or kernel executes it, the same `.spl` file can be retargeted across backends without modification — the capability C2 develops next.

**C2. The DODA principle.** Design Once, Deploy Anywhere: a single `.spl` specification is the invariant. The choice of model provider (Ollama, OpenRouter, Anthropic, Momagrid) and verifier (SymPy, SageMath, Lean) is resolved at invocation time via CLI flags, not embedded in the specification. The same file runs on a developer's laptop and on a distributed inference grid without modification — declarative composition is the mechanism that makes this possible.

**C3. Controlled empirical evaluation.** A 1200-run experiment (10 models × 20 problems × 2 arms × 3 repetitions; problems span 6 difficulty tiers) on symbolic mathematics provides quantitative evidence of the two-mode composition benefit and surfaces a structured capability hierarchy: models that score 100% in the probabilistic arm (LLM-only math) may score as low as 32% in the two-mode arm (solver), not because they cannot solve the math but because they cannot produce the structured decomposition the deterministic engine requires. This reveals that format-compliance for declarative composition is a separable capability from mathematical reasoning.

**C4. The verifier ladder.** We implement a three-rung hierarchy of symbolic verifiers — SymPy (algebraic), SageMath (mathematical), Lean 4 (formal proof) — as ascending correctness guarantees composable into any SPL workflow via `ASSERT`. R1 and R2 are evaluated empirically in the 1200-run experiment (C3); R3 (Lean 4) is implemented with a working bridge (15/15 unit tests passing) and demonstrated as a design contribution, scoped outside this paper's empirical evaluation. Each rung is a Python-callable tool registered at runtime; the workflow specification is agnostic to which rung is active. The ladder is itself declared, not coded: the escalation logic is `ASSERT` chains in `.spl`, not Python conditionals.

### 1.4 Paper Organization

Section 2 surveys related work, positioning SPL against imperative frameworks, declarative systems, and neurosymbolic AI. Section 3 presents the SPL language design — the full primitive set, semantics, and formal grammar. Section 4 formalizes the verifier ladder. Section 5 describes the implementation: how the two-mode executor dispatches probabilistic nodes to the adapter layer and deterministic nodes to the kernel layer, with model provider and verifier both selected at invocation time. Section 6 presents the 1200-run empirical evaluation (r=3). Section 7 discusses strengths, limitations, and future directions. Section 8 concludes.

## 2 Related Work

### 2.1 The Evolution of Data Programming

The history of data programming offers a clear precedent for the transition SPL proposes. In the 1970s, IBM's System R project showed that a declarative query language (SQL) could match or exceed the performance of hand-coded navigational database access, because the declarative form exposed optimization opportunities invisible in imperative code [3]. By the 1980s, SQL had become the industry standard, and Oracle's PL/SQL extension showed that procedural control flow (loops, variables, exception handling) could be layered on top of declarative queries without sacrificing the optimizer's ability to reason about individual `SELECT` statements [4].

The parallel to LLM programming is direct. Today's imperative frameworks — LangGraph's state graphs, AutoGen's conversation loops, CrewAI's role-based agents — are the equivalent of pre-SQL navigational code: powerful, flexible, and impossibly fragmented. SPL proposes that the same declarative revolution can occur for LLM orchestration, following the same evolutionary arc: atomic declarative queries (`PROMPT`) compose into procedural workflows (`WORKFLOW`) that an optimizer can eventually rewrite and route without altering the specification.

A critical difference from the SQL era: data queries are deterministic. LLM orchestration is not. This demands new primitives — `EVALUATE` for semantic branching, `WHILE` for quality-gated iteration, `EXCEPTION` for probabilistic failure modes — that have no SQL equivalent. And it demands a second computation mode entirely: `SOLVE` and `ASSERT` for the cases where a result must be exact rather than approximate.

## 2.2 Imperative Orchestration Frameworks

**AutoGen** [1] focuses on multi-agent conversation patterns where agents exchange messages. Its strength is natural dialogue flows; its limitation is that orchestration logic is embedded in Python and tightly coupled to the runtime.

**CrewAI** [2] introduces a role-based metaphor where specialized agents collaborate. It is intuitive for team-like workflows, but remains Python-first with no path to compilation or optimization.

**PydanticAI** takes a code-first approach, prioritizing developer ergonomics and type safety. Like the above, it operates exclusively in the probabilistic mode with no symbolic integration layer.

None of these frameworks provides a language-level mechanism to say "execute this step deterministically and verify the result before the workflow proceeds."

## 2.3 Declarative LLM Systems

**LMQL** [5] pioneered SQL-like LLM interaction with output constraints (type, length, regex), achieving 26–85% cost reduction through constrained decoding. LMQL operates at the query level — individual prompts with output validation — and does not provide constructs for multi-step workflows or deterministic integration.

**DSPy** [6] introduces declarative modules (`Predict`, `ChainOfThought`) with a compiler that automatically optimizes prompts via few-shot bootstrapping. DSPy's "declarative" is at the module-interface level: it optimizes which prompts to use, not how to orchestrate multi-step workflows. It provides no iteration, no exception handling, no symbolic integration, and no formal grammar. A worked contrast showing how the same self-refine task is expressed in DSPy assertions and LMQL token constraints versus SPL's `EVALUATE`/`WHILE` runtime semantics is given in Appendix J.

**SGLang** [7] provides structured generation with runtime optimizations (RadixAttention, up to $6.4\times$ throughput). SGLang optimizes inference execution below the level of agentic patterns; it does not address workflow orchestration.

**Agent Spec** [8] defines a YAML/JSON schema for framework-agnostic agent specification. It is a configuration format, not an executable language, and lacks control flow, iteration, and exception handling within the specification itself.

**eBay DSL** [9] separates workflow specification from implementation across multiple backend languages, achieving 60% reduction in development time. It focuses on enterprise pipeline configuration and does not formalize LLM-specific primitives or symbolic integration.

**Agentics 2.0** [10] introduces a logical transduction algebra where LLM calls are typed semantic transformations composable via algebraically grounded operators. It is the closest concurrent work in formalism, though algebraic rather than SQL-inspired, and it still operates exclusively in the probabilistic mode.

**Compiled AI** [11] addresses non-determinism through compile-time LLM invocation: the language model runs once at specification time to generate a deterministic executable artifact; no LLM is invoked at runtime. This achieves strong execution guarantees at the cost of losing all runtime adaptability — no `EVALUATE` branching, no `WHILE` quality gates, no `EXCEPTION` recovery. SPL takes the opposite stance: both modes remain live and equally accessible at runtime, interleaved through a single declarative specification. The tradeoff is explicit: SPL retains full probabilistic expressiveness (multi-shot LLM calls, adaptive routing, exception handling) alongside deterministic verification.

**Blueprint First** [12] proposes an engineering pattern that decouples workflow logic from the generative model. Expert-defined procedures are codified into an "Execution Blueprint" (deterministic engine), and the LLM is invoked only for bounded sub-tasks within it. While this echoes SPL's DODA principle — the blueprint as invariant, the LLM as pluggable component — the approach remains a design pattern rather than an executable language. Blueprint First offers no symbolic verification layer (no `SOLVE`/`ASSERT` equivalent), no formal grammar, and no shared variable namespace allowing seamless data flow between modes. The blueprint is a manually maintained engineering artifact; SPL's specification is the executable, compiled form.

**PDL** [27] is a YAML-based prompt format with Jinja2 variable templating and 15 block types (`model:`, `code:`, `for:`, `repeat:`, etc.). PDL's design goal is structured, repeatable prompting: blocks render text in a document-oriented fashion, and `code:` blocks can invoke arbitrary Python. PDL does not define typed workflow variables, semantic branching conditioned on LLM output, formal exception handling, or a SOLVE/ASSERT verification layer; its evaluation consists of three qualitative case studies with no quantitative benchmark. SPL differs at the abstraction level: the unit of composition is a named, parameterized `WORKFLOW` with typed `INPUT/OUTPUT` declarations and first-class state persistence across CALL boundaries, not a document template. The two systems are complementary rather than competing: PDL excels at structured prompt authoring; SPL addresses multi-step workflow orchestration with formal correctness gates.

**APPL** [28] introduces a Python-embedded DSL for LLM programming that preserves Python control flow while providing prompt-construction helpers. APPL's integration with Python is tighter than SPL's (SPL is a separate language), giving more expressive power for ad-hoc scripting, but it provides no formal grammar, no ASSERT-based verification gate, and no symbolic solver integration.

**VIEIRA** [29] proposes a probabilistic-relational framework for programming with foundation models. Vieira treats language and vision models as stateless relational functions embedded in a Scallop datalog program; the logic substrate governs neuro-symbolic composition across language, vision, and database modalities. Unlike SPL, Vieira has no imperative workflow layer (no named procedures, WHILE repair loops, or CALL boundaries), no typed `INPUT/OUTPUT` declarations, and no hard `ASSERT` gate — its correctness mechanism is probabilistic relational inference rather than binary pass/fail verification.

## 2.4 LLM+Solver Composition

A distinct line of work explores coupling language models with symbolic solvers via mode composition: the LLM translates a natural language problem into a solver encoding, the solver executes deterministically, and the LLM translates the result back to natural language.

**LLM+P** [13] pioneered this pattern: language models generate PDDL planning encodings; a classical optimal planner (Fast Downward) solves them; the LLM narrates the solution. This foundational work established the "two-phase" pattern — probabilistic translation, deterministic execution, probabilistic interpretation — that has influenced subsequent systems. The limitation is that the LLM is restricted to a translator role between fixed input/output schemas (NL $\leftrightarrow$ PDDL); there is no multi-step workflow composition, shared variable namespace, or first-class language for orchestrating the two phases.

**MCP-Solver** [14] wraps constraint solvers (MiniZinc, PySAT, Z3, Clingo/ASP) as Model Context Protocol tools. The LLM builds a solver encoding through conversational tool calls, the solver executes, and the LLM interprets results. Like LLM+P, this uses a *tool-call pattern*: the LLM invokes the solver as a black box via a protocol-defined schema. There is no formal workflow language, no explicit mode boundary visible in source, and the variable-passing contract is implicit in tool schemas rather than explicit in a grammar.

**DUPLEX** [15] applies dual-system composition to robotic task planning: a lightweight "fast" system maps entity relations to PDDL via LLM; a classical planner executes; a "slow" system (heavier LLM with solver diagnostics) activates on failure. DUPLEX exhibits the PLAN $\rightarrow$ SOLVE $\rightarrow$ ASSERT $\rightarrow$ EXPLAIN lifecycle that SPL formalizes as a language pattern. However, DUPLEX is confined to the PDDL domain; modes do not share variables — they communicate through rigid schema — and the composition is hard-coded to the two-phase structure, not visible as a declarative grammar.

Across all three systems, the LLM is a component *within* a larger orchestration pattern, not a co-equal participant with explicit mode boundaries visible in source. SPL differs fundamentally: `SOLVE` and `ASSERT` are language primitives (not tool calls), the mode boundary is visible in the grammar, and variables flow between modes through a unified `@variable` namespace without schema marshaling. In a tool-call system, the developer writes explicit serialization/deserialization logic in Python to pass an LLM's string output to a SymPy tool and parse the result back; in SPL, the `{@variable}` interpolation syntax makes this a language-level feature managed by the runtime.

## 2.5 Neurosymbolic AI

A growing body of work combines neural and symbolic computation, but at the model or algorithm level rather than the workflow language level.

**AlphaGeometry** [16] solves Olympiad geometry problems by alternating a neural language model (generating proof steps) with a symbolic deduction engine (verifying them). The neuro-symbolic loop is hard-coded to the geometry domain and implemented as a custom Python pipeline. SPL generalizes this pattern to arbitrary domains: the `SOLVE`/`ASSERT` primitives express any such loop declaratively, and the verifier (geometry engine, computer-algebra system, or any other symbolic tool) is a pluggable runtime choice rather than a hard-coded dependency.

**Neural theorem proving** systems (PACT, LeanDojo [17], HyperTree Proof Search [18]) use LLMs to generate proof tactics for a formal proof assistant, with the assistant itself as the verifier. SPL takes a different approach at the language level: rather than a fixed model-plus-proof-assistant pipeline, any symbolic verifier — a proof assistant included — is a pluggable `ASSERT` target selected at runtime, not hard-coded into the language.

**Program synthesis** systems (Codex [19], AlphaCode [20]) generate code and execute it to verify functional correctness. SPL's `SOLVE` construct is the declarative equivalent of this loop: the programmer specifies the expression template; the kernel executes it; `ASSERT` verifies the result — without the programmer writing the execution and verification harness in Python.

Recent work on composable neuro-symbolic architectures offers insights into the two-mode approach. **Compositional AI Beyond LLMs** [23] examines from a systems/hardware perspective the characteristics of neuro-symbolic-probabilistic architectures, finding distinct memory and compute profiles and consistent performance gains over monolithic LLMs of comparable size. **Symbolic Seams** [24] proposes "symbolic seams" — explicit architectural breakpoints with typed boundary objects — as a design principle for composable systems, conceptually aligned with the mode boundary that SPL makes visible at the language level.

The common thread across these systems is that neuro-symbolic integration is implemented as bespoke application logic rather than as a language primitive. SPL's contribution is to make that integration a first-class, reusable construct expressible in few keywords.

## 2.6 Positioning Summary

SPL is the only system with `Yes` across all four differentiating columns in Table 1: (1) *Grammar* — both computation modes have first-class status in the formal EBNF (Appendix A), not just documented convention; (2) *Semantic Eval* — `EVALUATE` provides native LLM-as-judge branching on natural-language conditions; (3) *Both modes @runtime* — adaptive control flow (`EVALUATE`, `WHILE`, `EXCEPTION`) operates across both modes during execution, not only at specification time; (4) *Sym. Integration* — `SOLVE`/`ASSERT` provide native, pluggable dispatch to a symbolic solver, not hand-written Python glue. A fifth property that Table 1 does not tabulate as a column, because no comparison system offers a partial case worth contrasting, is SPL's unified `@variable` namespace: outputs of one mode are available as inputs to the other without tool-schema marshaling. The critical distinction from every other declarative system is that SPL's declarativity extends across the mode boundary: the `.spl` source specifies composition of both modes; no Python glue bridges them.

**Table 1: System Comparison.** Column definitions: *Grammar* — both computation modes (if the system has more than one) have first-class status in a formal grammar, not merely a documented convention; *Semantic Eval* — native LLM-as-judge branching on natural-language conditions (SPL's `EVALUATE`), as opposed to string/regex matching only; *Both modes @runtime* — adaptive control flow (branching, iteration, exception handling) can operate across both modes during execution, not only at specification time; *Sym. Integration* — native, pluggable dispatch to a symbolic solver or verifier, as opposed to hand-written Python glue connecting the LLM layer to an external tool.

| System | Paradigm | Grammar | Semantic Eval | Both modes @runtime | Sym. Integration |
|---|---|---|---|---|---|
| LangGraph | Imperative | No | No | No | Manual (Python) |
| AutoGen | Imperative | No | No | No | Manual (Python) |
| CrewAI | Imperative | No | No | No | Manual (Python) |
| LMQL | Declarative | Yes | No | No | No |
| DSPy | Declarative | No | No | No | No |
| SGLang | Declarative | Yes | No | No | No |
| LLM+P | Two-phase | No | No | No | Partial (PDDL) |
| Blueprint First | Declarative | No | No | No | No |
| Compiled AI | Declarative | No | No | No | No |
| MCP-Solver | Tool-call | No | No | Yes (ad hoc) | Via protocol |
| DUPLEX | Two-phase | No | No | No | Yes (PDDL domain) |
| Agent Spec | Declarative | Yes (schema) | No | No | No |
| eBay DSL | Declarative | Yes | No | Yes | No |
| Agentics 2.0 | Algebraic | Yes (algebra) | No | No | No |
| AlphaGeometry | Neuro-sym. | No | No | Yes (hard-coded) | Yes (domain-specific) |
| **SPL** | **Declarative** | **Yes (EBNF)** | **Yes** | **Yes (first-class)** | **Yes (pluggable)** |

## 3 The SPL Language

### 3.1 Design Principles

SPL is governed by the following design principles:

**P1. Declarative composition over imperative implementation.** The programmer specifies *what* each step computes and *which mode* it inhabits — not which model, which verifier, or which execution path achieves it. The two modes may be woven loosely (distinct probabilistic and deterministic phases passing data between them) or tightly (fine-grained interleaving where `ASSERT` failure immediately re-enters `GENERATE`); SPL handles both ends of this spectrum with the same primitives. The runtime resolves the mode boundary; the `.spl` source is the invariant across both coupling styles, both model providers, and both deployment environments.

**P2. Two-mode primitives as co-equal first-class constructs.** SPL provides primitives for both computation modes — System 1 (probabilistic, intuitive) and System 2 (deterministic, mechanical) in the dual-process sense [21], reinterpreted epistemically rather than temporally (see §1.2). The probabilistic mode (`GENERATE`, `EVALUATE`, `WHILE`, `EXCEPTION`) addresses the characteristics of LLM computation: approximate, non-reproducible, quality-assessed, cost-sensitive. The deterministic mode (`SOLVE`, `ASSERT`) addresses the characteristics of symbolic computation: exact, reproducible, machine-verifiable. Neither mode is subordi-

nate to the other in the language design — their composition in a single specification is the defining property of SPL.

**P3. Shared variable space.** Outputs of `GENERATE` steps are available as inputs to `SOLVE` steps, and vice versa, through a single `@variable` namespace. The programmer does not marshal values between modes; the runtime manages the boundary. Two syntactic forms govern variable use: `@name TYPE` is the *declaration/binding* form (used in `INPUT`/`OUTPUT` signatures, `SOLVE` targets, and assignment), while `{@name}` is the *interpolation/evaluation* form (used inside `PROMPT` templates, `SOLVE` expressions, and `ASSERT` predicates). The executor substitutes each `{@name}` reference with the current value of `@name` before dispatch. This is the mode-crossing seam, and it is the same mechanism as SPL f-strings (`f"..."`) applied uniformly across all string contexts.

**P4. DODA — Design Once, Deploy Anywhere.** The `.spl` file is the invariant. Physical decisions — which LLM, which verifier, which infrastructure — are resolved at invocation time via `--adapter`, `--model`, and `--kernel` flags. The same specification is the logical description of the workflow regardless of where it runs.

**P5. Compilation target.** The language is designed to produce an optimizable intermediate representation (IR), enabling future optimizer passes (GENERATE merging, SELECT caching, model routing) without changing the surface language.

## 3.2 The SQL Analogy

SPL's declarative form is not an independently-argued virtue borrowed from SQL; it is the mechanism that makes DODA (C2, §1.3) possible. A specification that declares only *what* each step computes, not *which* model or kernel executes it, can be retargeted across backends without modification — demonstrated directly by the same unmodified `.spl` recipe running to closely-agreeing outcomes on two independently-implemented runtimes (§5). The SQL and PL/SQL stack below is offered only as a mental-model aid for reading SPL's syntax, not as the argument for why declarative form matters here:

- `SELECT` assembles the context that flows *into* the LLM — analogous to `SELECT ... FROM ... WHERE` gathering rows from a table.
- `GENERATE` invokes the LLM and captures what flows *out* — analogous to the result set returned by a SQL query.
- `PROMPT` binds `SELECT` and `GENERATE` into a named, independently executable unit — analogous to a single SQL query.
- `WORKFLOW` adds variables, loops, branching, and exception handling around `PROMPT` calls — analogous to PL/SQL layering procedural control flow over SQL queries.
- `SOLVE` routes a Python expression to the deterministic kernel — an extension that has no SQL analogue.

This last point marks the boundary of the SQL analogy. SPL must go further because LLM orchestration requires a second computation mode that relational data access does not. A second boundary deserves acknowledgment: SQL optimization rests on relational algebra equivalences over deterministic set operations, whereas `GENERATE` samples from a probability distribution. Future SPL optimizations (batching, caching, model routing) must therefore operate at the *workflow* level — reordering independent nodes, fusing adjacent calls — rather than applying algebraic rewrites to individual `GENERATE` invocations.

## 3.3 Deterministic Primitives

### 3.3.1 SOLVE — Kernel-Routed Computation

`SOLVE` dispatches a computation to the selected verifier engine and assigns the result to a workflow variable. Its behavior is backend-dependent: for the SymPy and SageMath rungs (R1/R2), it evaluates a Python expression in the live IPython kernel, as shown below; for the Lean 4 rung (R3), it instead drives a multi-stage formalize/typecheck/judge-faithfulness/prove protocol rather than evaluating a single expression. The

surface syntax is unchanged across rungs, but the underlying computation model differs by rung; see §4.4 for the Lean case.

```
SOLVE @derivative SYMPY := "diff({@expression}, x)"
```

The `{@variable}` syntax binds any `@variable` currently in scope into the expression before dispatch. This is the boundary-crossing mechanism: the LLM's output (held in `@expression`) becomes the input to the symbolic engine without programmer-managed marshaling. The result (`@derivative`) is available immediately to subsequent `GENERATE` or `SOLVE` steps. The substitution is performed by the executor via regex matching on the parsed token stream — the runtime resolves each `{@identifier}` reference against the variable store before constructing the kernel expression. The `SOLVE` kernel (SymPy/SageMath/Lean) runs in an out-of-process IPython subprocess; `CREATE TOOL_API` code executes via `exec()` within an isolated in-process namespace containing only the declared parameters and Python stdlib. Malformed expressions raise a `SyntaxError` or `NameError` caught by the `EXCEPTION` handler.

`SOLVE` is mode-explicit by design. The programmer cannot accidentally invoke the symbolic engine from a `GENERATE` call or the LLM from a `SOLVE` call. The mode boundary is visible in the source.

### 3.3.2   ASSERT — Verification Gate

`ASSERT` evaluates a Boolean predicate and gates workflow continuation on the result. The executor dispatches to one of two paths depending on whether a kernel session is active: (1) the IPython kernel, for arbitrary Python expressions that require imports or a live computation environment (e.g., symbolic math, graph libraries); (2) the registered `TOOL_API` namespace, for predicates that call a registered tool directly — these execute without a kernel and require no `-kernel` flag:

```
-- Kernel path: symbolic math expression (requires --kernel)
ASSERT simplify({@sympy_result} - diff({@expression}, x)) == 0
  OTHERWISE RETURN @explanation WITH status = 'verification_failed'

-- Tool-API path: registered TOOL_API predicate (no kernel required)
ASSERT is_optimal(@solution)
  OTHERWISE RETURN @report WITH status = 'infeasible'
```

On success, the workflow continues — the claim is machine-verified. On failure, the `OTHERWISE` body executes, typically committing a result with an error status or escalating to a higher verifier rung. `ASSERT` is the mechanism by which SPL workflows achieve correctness guarantees: a step that passes `ASSERT` is proven correct by the deterministic predicate, not merely judged plausible.

It is important to distinguish two failure modes that `ASSERT` separates: (a) *execution failure* — the generated symbolic plan raises a `SyntaxError` or `NameError` in the kernel, indicating the LLM produced structurally invalid code; and (b) *semantic failure* — the plan executes without error but `ASSERT` returns false, indicating the plan encodes an incorrect transformation (e.g., the derivative of a wrong expression). The solver-arm pass criterion in §6 counts a cell as passing only when the kernel executes all steps *and* the final `ASSERT verify({@result}, {@problem})` returns true. Execution success alone is not a pass. This distinction is the core of what "machine-verified correctness" means in this paper: the ASSERT predicate is the ground-truth oracle, not a proxy for successful execution. Because `ASSERT` accepts any Python Boolean expression, tolerance-based assertions for numerical computation are naturally supported (e.g., `ASSERT abs({@result} - {@expected}) < 1e-6`); the language imposes no restriction to exact symbolic equality.

The deterministic primitives are not limited to mathematics. Any Python-evaluable predicate works as an `ASSERT` condition:

```
-- Schema validation: verify LLM output conforms to expected JSON structure
ASSERT json.loads({@api_response}) and validate(json.loads({@api_response}), schema)
  OTHERWISE RETURN @result WITH status = 'schema_violation'
```

```
-- Length constraint: ensure generated content meets minimum requirements
ASSERT len({@report_body}) >= 500
  OTHERWISE RETRY

-- Graph property: verify generated network satisfies connectivity invariant
ASSERT nx.is_connected(G) and nx.number_of_nodes(G) == {@expected_nodes}
  OTHERWISE RETURN @graph WITH status = 'graph_invalid'
```

These examples illustrate that `SOLVE`/`ASSERT` form a general *deterministic computation mode*, not a math-specific feature. `SOLVE` always routes to the IPython kernel. `ASSERT` dispatches to the kernel for arbitrary Python expressions, or directly to the registered tool namespace when the predicate is a `TOOL_API` call — enabling formal gates without a kernel session. Any domain with a Python-callable verifier — JSON schema validation, unit checking (pint), constraint solving (Z3), statistical testing (scipy), LP/MIP solvers (PuLP) — plugs into the same primitives. We label this mode "deterministic" throughout the paper to emphasize its reproducibility and machine-verifiability relative to LLM inference. Strictly, the Python kernel can execute non-deterministic code (e.g., MCMC sampling, randomized algorithms); in such cases the `ASSERT` gate still enforces a verifiable postcondition on the result, preserving the mode's role as the verification boundary regardless of whether the kernel computation itself is stochastic.

### 3.3.3  CALL PARALLEL — Concurrent Branch Dispatch

```
CALL PARALLEL
  CALL arm_solver(@problem) INTO @result_solver
  CALL arm_llm_only(@problem) INTO @result_llm
END
```

Branches execute concurrently via `asyncio.gather` on a single node, or route to distinct Momagrid worker nodes on a distributed grid. This construct makes A/B experiments — deterministic arm vs. probabilistic arm — a language-level primitive rather than a test-harness concern. When parallel branches contain `SOLVE` or `ASSERT` steps, each branch operates on an isolated copy of the kernel namespace to prevent race conditions; results are merged into the parent scope only upon branch completion via the `INTO` binding.

### 3.4  Probabilistic Primitives

### 3.4.1  SELECT and GENERATE — The Atomic I/O Contract

`SELECT` assembles the prompt from multiple context sources; `GENERATE` invokes the LLM and captures the result:

```
WORKFLOW explain_concept
  INPUT: @concept TEXT, @audience TEXT DEFAULT 'undergraduate'
  OUTPUT: @explanation TEXT
DO
  GENERATE explanation(concept, audience) INTO @explanation
    SELECT
      system_role('You are a clear, precise instructor.'),
      @concept AS concept,
      @audience AS audience
    PROMPT "Explain {@concept} to a {@audience} student in 3 sentences."
END
```

In the common case shown above, each `AS` clause aliases a variable to itself (`@concept AS concept`) — `SELECT`'s load-bearing job here is assembling heterogeneous context sources (`system_role(...)` alongside workflow variables) into a single named binding set for `PROMPT` interpolation, not renaming. `AS` earns its keep when a source and its target name genuinely differ (e.g., selecting from a computed sub-expression or

a value bound under a different name inside a `CALL` frame), but that case is not exercised in this listing and we do not claim otherwise. `SELECT/AS` was introduced in SPL v1.0, before `CREATE FUNCTION` (Appendix A) existed as a distinct construct for named, parameterized, reusable prompt templates; for the common case of explicit prompt engineering, `CREATE FUNCTION` is now the more direct mechanism, and `SELECT/AS` remains in the language for inline, one-off context assembly rather than as the primary prompt-engineering surface.

### 3.4.2 EVALUATE — Semantic Branching

`EVALUATE` uses the LLM as a runtime judge for conditions that cannot be expressed as deterministic comparisons:

```
EVALUATE @explanation
  WHEN contains('undefined') OR contains('undefined symbol') THEN
    @explanation := @explanation + '\n\n[Note: terms defined in appendix]'
  WHEN 'the explanation is unclear or contains unexplained jargon' THEN
    @status := 'needs_revision'
  WHEN = 'satisfactory' THEN
    @status := 'complete'
  ELSE
    @status := 'needs_revision'
END
```

Three condition forms coexist here, each dispatched differently: `contains(...)` is a deterministic string-containment check; the free-form quoted string `'the explanation is unclear or contains unexplained jargon'` is a natural-language predicate with no deterministic reduction, dispatched to the LLM judge — this is the branch that demonstrates genuine semantic evaluation, since no exact-match or substring test could implement it; and `= 'satisfactory'` is a literal equality comparison, evaluated deterministically against a prior exact-match value (e.g., a status field set by an earlier step), not against `@explanation`'s free text. This hybrid detection rule means the programmer does not annotate which branches are semantic and which are deterministic — the parser infers it from syntax: quoted natural-language conditions go to the judge, containment/equality checks are evaluated directly.

### 3.4.3 WHILE — Quality-Gated Iteration

```
WHILE 'explanation is incomplete or unclear' DO
  GENERATE refine_explanation(@explanation, @feedback) INTO @explanation
END
```

The termination condition is evaluated by the LLM on each iteration: the executor assembles all current `@variable` values as context and asks the LLM judge whether the condition still holds. A built-in maximum iteration guard (configurable, default 15) prevents infinite loops.

### 3.4.4 EXCEPTION — LLM-Aware Error Handling

```
EXCEPTION
  WHEN HallucinationDetected THEN
    RETURN @result WITH status = 'hallucination_detected'
  WHEN ContextLengthExceeded THEN
    @problem := summarize(@problem)
    RETRY
  WHEN BudgetExceeded THEN
    RETURN @partial_result WITH status = 'budget_exceeded'
END
```

The exception taxonomy formalizes failure modes specific to LLM computation — hallucination, refusal, context overflow, budget violation — that have no equivalent in conventional exception hierarchies. Detection

mechanisms vary by type: `ContextLengthExceeded` and `BudgetExceeded` are raised deterministically by the adapter when API limits are hit; `ModelRefused` is detected by pattern matching on refusal templates in the response. `HallucinationDetected` is currently a proposed exception type — the runtime does not yet implement automatic hallucination detection; the programmer can raise it explicitly via an `EVALUATE` guard that dispatches to an LLM judge. The taxonomy is designed to be extensible as detection capabilities mature.

### 3.5 The Two-Mode Composition Pattern

Real agentic tasks follow a recurring four-stage lifecycle: **PLAN** (System 1 decomposes the problem), **SOLVE** (System 2 executes it exactly), **ASSERT** (System 2 verifies the result), **EXPLAIN** (System 1 narrates it in natural language). The following example shows this lifecycle expressed declaratively in the following SPL workflow:

```
WORKFLOW solve_with_verification
  INPUT: @problem TEXT
  OUTPUT: @explanation TEXT
DO
  -- [PLAN -- System 1 / Probabilistic] LLM decomposes the problem
  GENERATE decomposition(@problem) INTO @steps_text
    SELECT @problem AS problem
    PROMPT 'Decompose this problem into symbolic computation steps,
            one per line in the format: expression|operation'

  -- [PLAN -- System 1 / Probabilistic] Sanity gate before kernel handoff
  GENERATE plan_check(@steps_text, @problem) INTO @plan_verdict
    SELECT @steps_text AS plan, @problem AS problem
    PROMPT 'Does this plan correctly address the problem? Reply: pass or fail'

  EVALUATE @plan_verdict
    WHEN = 'fail' THEN
      RETURN @explanation WITH status = 'plan_sanity_error'
  END

  -- [SOLVE -- System 2 / Deterministic] Kernel executes each symbolic step exactly
  CALL solve_chain(@steps_text) INTO @verified_result

  -- [ASSERT -- System 2 / Deterministic] Verify the chain answer against the problem
  ASSERT verify({@verified_result}, {@problem})
    OTHERWISE RETURN @explanation WITH status = 'verification_failed'

  -- [EXPLAIN -- System 1 / Probabilistic] LLM narrates the verified result
  GENERATE explanation(@verified_result, @problem) INTO @explanation
    SELECT @verified_result AS result, @problem AS problem
    PROMPT "Explain this verified solution in clear prose: {@result}"

  RETURN @explanation WITH status = 'complete'
END
```

This 30-line workflow composes five mode transitions declaratively: System 1 decomposition → System 1 sanity gate → System 2 execution → System 2 verification → System 1 narration. Each transition is expressed in SPL syntax; no Python bridges the modes. (Both `solve_chain()` on the `CALL` line and `verify()` on the `ASSERT` line are user-registered Python callables — the former dispatching the decomposed steps to the symbolic kernel, the latter checking the result — not built-in SPL primitives; see Appendix I for implementation patterns and a source-code reference for both.)

The `{@var}` template syntax is the *formalization boundary* of SPL: the seam at which System 1 outputs (unstructured text from `GENERATE`) are cast into typed expressions consumed by System 2 (`SOLVE/ASSERT`). The programmer does not write serialization code; the runtime resolves variable bindings before kernel dispatch. The mode crossing is visible at the source level — every `{@...}` reference marks a System 1 → System 2 handoff — and is managed entirely by the language.

The programmer writes the same `.spl` file regardless of which LLM and which symbolic engine execute at runtime — the declarative specification is the invariant, and the DODA principle holds across both the model axis and the verifier axis.

### 3.6 Formal Grammar

The SPL grammar defines the following key production rules for two-mode integration:

```
solve_stmt    ::= 'SOLVE' '@'identifier ['SYMPY' | 'SAGE' | 'LEAN'] ':=' string_expr
assert_stmt   ::= 'ASSERT' python_expr ['OTHERWISE' stmt_block]
parallel_stmt ::= 'CALL' 'PARALLEL' (call_stmt)+ 'END'
stmt          ::= ... | solve_stmt | assert_stmt | parallel_stmt
```

The `SYMPY | SAGE | LEAN` type annotation on `SOLVE` is optional; when omitted, the runtime selects the kernel registered at invocation time. The full grammar is provided in Appendix A.

## 4 The Verifier Ladder

### 4.1 Motivation: Ascending Certification Scope

LLM-generated mathematical reasoning is probabilistic by nature: a model scoring 95% on a benchmark is wrong on 1 in 20 problems, with no signal to the caller about which ones. Symbolic verification closes this gap — but symbolic verification itself is not monolithic. A SymPy check of polynomial differentiation takes milliseconds and requires no external tooling. A formal Lean 4 proof of the same claim takes seconds, requires Mathlib, and produces a machine-checkable certificate that any Lean installation can re-verify independently. The two checks are not interchangeable: one is appropriate for routine computation, the other for publication-quality assertions. **The rungs differ in what class of claim each can certify, not in how trustworthy the engine is within that class**: a SymPy identity check and a Lean proof are each exactly correct for the claims they are built to check — escalating a rung is a decision to seek a stronger form of certificate, not a correction of a less reliable one.

The SPL verifier ladder makes this trade-off explicit and programmable. Three rungs address different cost/confidence operating points, ordered by the scope of what they certify rather than by reliability:

| Rung | Engine | What it certifies | Setup | Typical use |
|---|---|---|---|---|
| R1 | SymPy (`python3` kernel) | Algebraic identity, calculus, matrix operations | `pip install sympy` | Routine STEM computation |
| R2 | SageMath (`sagemath` kernel) | Number theory, geometry, polynomial rings, combinatorics | `pip install 'spl-llm[sage]'` | Deeper mathematical claims |
| R3 | Lean 4 (`lean_bridge`) | Machine-checked formal proof against Mathlib | `bash setup_lean.sh --with-mathlib` | Publication-quality assertions |

A workflow selects a rung via the `--kernel` and `--param backend=` flags at invocation time, optionally probing multiple rungs in parallel via `CALL PARALLEL`, with each rung receiving an engine-appropriate expression for the same underlying claim, to find the highest rung that certifies it. Expression syntax is rung-specific — a SymPy call, a Sage call, and a Lean statement are written differently for the same claim, as §4.5 shows — but the *workflow structure* (which rung to try, in what order, with what fallback) is identical across all

three rungs. The DODA principle applies to verifier selection just as it applies to model selection: the orchestration logic, not the expression text, is what stays invariant.

## 4.2 Rung R1: SymPy

The SymPy rung runs inside an IPython kernel started with the standard `python3` kernelspec. Seventeen mathematical operations are registered as SPL tools via `CREATE TOOL_API`:

```
CREATE TOOL_API solve_step_with_sympy(expression, operation)
  '''
  import sympy as sp
  from sympy.abc import x, y, z, t, n
  ops = {
    "diff":        lambda e: sp.diff(e, x),
    "integrate":   lambda e: sp.integrate(e, x),
    "solve":       lambda e: sp.solve(e, x),
    "eigenvalues": lambda e: sp.Matrix(eval(e)).eigenvals(),
    "dsolve":      lambda e: sp.dsolve(sp.sympify(e)),
    ...   -- 12 further operations
  }
  result = ops[operation](sp.sympify(expression))
  return f"{result}|{sp.latex(result)}"
  '''
```

The tool returns a `bare_result|human_readable` protocol: the bare result is available as a string for subsequent `{@var}` interpolation; the human-readable side is written to the chain trace log. On failure the sentinel `solver_error|...` is returned, which the workflow detects via `EVALUATE` and exits with an explicit status rather than propagating a bad value downstream.

## 4.3 Rung R2: SageMath

SageMath extends SymPy's algebraic reach into number theory (Galois groups, elliptic curves, modular arithmetic), differential geometry, and combinatorics. The runtime discovers the SageMath Jupyter kernelspec automatically and raises a clear install error if absent (see Appendix C for setup). The same `SOLVE / ASSERT` primitives and `{@var}` interpolation work unchanged — the only runtime difference is which kernelspec launches. This is the DODA principle applied at the verifier level: the workflow programmer writes identical SPL regardless of which mathematical engine executes beneath.

## 4.4 Rung R3: Lean 4

The Lean rung targets a qualitatively different class of claim: a machine-checked formal proof against Mathlib rather than an algebraic identity, via a four-stage formalize/typecheck/judge-faithfulness/prove protocol (full SPL source in Appendix G.5) that returns a *badge*: `machine_proved`, `statement_checked`, `unfaithful`, or `unverified`. The Lean kernel, not the LLM, certifies the Typecheck and Prove stages; a failed proof never blocks delivery, only withholds the higher badge, so the rung can run speculatively in a `CALL PARALLEL` branch alongside a SymPy arm that already delivers an answer.

One stage remains probabilistic: `judge_faithfulness`, a second LLM call comparing the formalized Lean statement against the original claim, is the sole non-kernel-certified gate in the protocol — typechecking and provability alone cannot catch a well-formed, provable Lean statement that still misrepresents what it was asked to formalize. `machine_proved` therefore certifies the proof *for the statement the judge accepted as faithful*, not that the judge was correct to accept it; this paper reports no independent accuracy estimate for that judge (extended discussion, including the design rationale and cost/setup trade-offs across rungs, in Appendix G.5). As stated in §1.3 (C4) and §6.1, R3 is a working design contribution (15/15 unit tests passing) and is not part of the 1,200-run empirical evaluation. One scoping note on DODA: the formalization

stage currently uses a general-purpose model through an ordinary `GENERATE` call, so the "same spec, any model" claim holds as implemented; were a fine-tuned NL→Lean translator introduced for this stage, it would be a fixed-translator configuration for that step specifically — outside the general DODA claim rather than a violation of it, the same way a hard-coded prompt template would be.

### 4.5 `ASSERT` as the Inter-Rung Gate

`ASSERT` is the mechanism by which the verifier ladder enforces correctness within a workflow. Each rung produces a value that can be checked, and `ASSERT` failure triggers escalation:

```
-- Try R1 first (inside a workflow body)
SOLVE @derivative SYMPY := "diff({@expression}, x)"
ASSERT simplify({@derivative} - diff({@expression}, x)) == 0
  OTHERWISE
    -- R1 could not verify; escalate to R2
    SOLVE @derivative SAGE := "SR({@expression}).diff(x)"
    ASSERT bool({@derivative} == diff({@expression}, x))
      OTHERWISE RETURN @explanation WITH status = 'r2_failed'
```

This escalation pattern is idiomatic SPL declarative composition: try the cheapest rung, gate on `ASSERT`, compose with a higher rung on failure. No Python glue code is required; the entire escalation logic — including the mode transitions between LLM narration and kernel verification — is expressed declaratively in the `.spl` source. The hand-written equivalent requires the programmer to manage kernel sessions, exceptions, and result marshaling explicitly:

```
sympy_kernel = start_kernel("python3")
try:
    derivative = sympy_kernel.execute(f"diff({expression}, x)")
    if not sympy_kernel.execute(
        f"simplify({derivative} - diff({expression}, x)) == 0"
    ):
        raise VerificationError("r1_failed")
except (KernelError, VerificationError):
    sage_kernel = start_kernel("sagemath")
    try:
        derivative = sage_kernel.execute(f"SR({expression}).diff(x)")
        if not sage_kernel.execute(
            f"bool({derivative} == diff({expression}, x))"
        ):
            return {"status": "r2_failed"}
    except KernelError:
        return {"status": "r2_failed"}
    finally:
        sage_kernel.close()
finally:
    sympy_kernel.close()
```

Two kernel lifecycles, two exception paths, and manual result marshaling between Python and each kernel's native representation replace the six-line `.spl` block above — the SPL runtime manages the kernel-session and exception-handling code implicitly, so line count is only part of what changes.

### 4.6 Pluggability

The verifier ladder is open-ended: any Python-callable verifier plugs in without modifying the language or runtime. A workflow author registers a new tool via `CREATE TOOL_API`, selects it via a parameter or `--kernel`

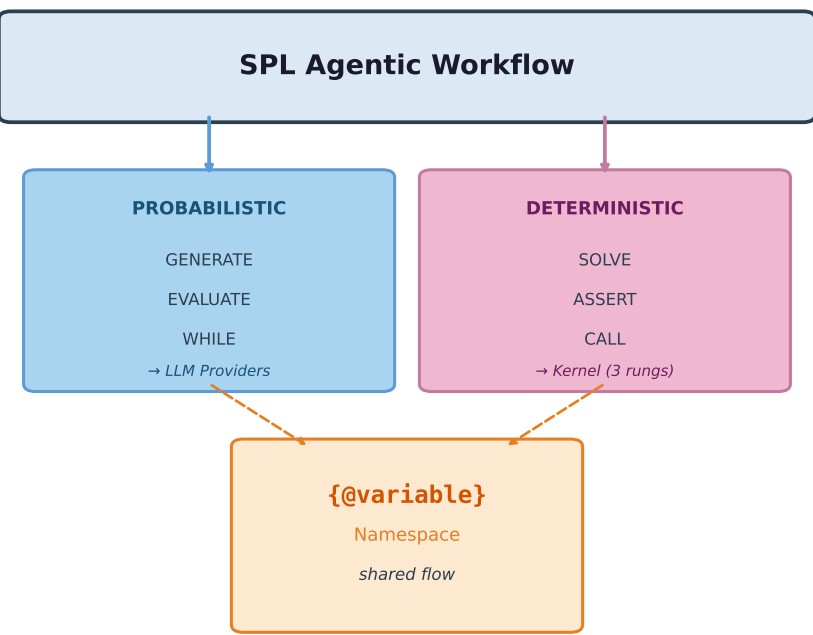

Figure 1: Execution Pipeline: The executor dispatches probabilistic nodes to the adapter layer (14 LLM providers) and deterministic nodes to a pluggable kernel, with both modes sharing state through the {@variable} namespace.

flag, and writes `ASSERT` conditions against its output. Beyond the three rungs demonstrated in this paper, the pattern generalizes to: unit verification (pint), graph property checking (NetworkX), constraint solving (Z3), type checking (mypy), or statistical hypothesis testing (scipy.stats). The language makes no assumption about what the kernel computes — only that it is Python-callable and that the result is a string the workflow can inspect.

## 5 Implementation

The executor parses a `.spl` source into an AST and dispatches each node to one of two engines: probabilistic nodes (`GENERATE, EVALUATE, WHILE, EXCEPTION`) route to the adapter layer, which normalizes calls across 14 LLM providers behind a two-method interface (`generate / generate_multimodal`). Deterministic nodes (`SOLVE, ASSERT`) route to the deterministic layer. `SOLVE` always dispatches to a persistent out-of-process Jupyter kernel for symbolic computation. `ASSERT` dispatches to the kernel when a kernel session is active; otherwise it evaluates the predicate directly in the registered `TOOL_API` namespace, enabling formal gates without a kernel session. The kernel activates only when `SOLVE` is encountered or `--kernel` is specified; otherwise no kernel process is started. Adapter and kernel are both selected at invocation time (`--adapter`, `--model`, `--kernel`) and never named in the `.spl` source — the deterministic mode of DODA mirrors the probabilistic one.

The execution pipeline routes `GENERATE/EVALUATE` to the adapter layer, `SOLVE` to the kernel, and `ASSERT` to either the kernel or the registered tool namespace depending on whether a kernel session is active, with all modes sharing state for seamless mode crossing without marshaling code.

The two modes share state through `{@var}` interpolation: before a `SOLVE` expression is dispatched to the kernel, the executor inlines current SPL variable values into the expression string. The result is captured and bound back to a `@var`, making it available to subsequent `GENERATE` or `EVALUATE` nodes. This variable-

passing contract is the integration seam between the two modes; no explicit marshalling code is required in the workflow.

The declarative form does more than offer syntactic convenience: it is designed to *enable* optimizations that are impossible in imperative orchestration code. A static optimizer is already implemented and shipped (`spl.optimizer.Optimizer`, exposed via `spl3 explain <file>.spl`): it walks the AST without making any LLM call and produces a per-`PROMPT`/`WORKFLOW` execution plan — token-budget allocation across `SELECT` sources, priority ordering, proportional compression under budget pressure, and cost estimation — which a hand-written Python orchestration script has no equivalent surface to analyze, because its LLM calls and control flow are interleaved in ordinary code rather than declared as data.

This static-planning capability is mirrored in the independent Go runtime (`https://github.com/digital-duck/SPL.go`, `internal/executor/planner.go`), giving two independent implementations agreement on what the IR means. The two runtimes are validated not only at this static-planning level but at full experimental scale: the identical `.spl` neurosymbolic recipe used for the results in §6 was run to completion on both — ten models, twenty problems, two solver modes, three repetitions, 1,200 cells each, no source changes between runtimes — yielding closely agreeing pass rates (87.2% Python vs. 87.9% Go) and round-trip-verified rates (88.3% vs. 88.8% of Pattern-1-pass runs).

What remains future work (§7.7) is *runtime* rather than static optimization: fusing adjacent `GENERATE` calls into a single batched API request, caching `SELECT` results across workflow runs, reordering independent `SOLVE` steps for kernel throughput, and routing cheap sub-tasks to smaller models — all without modifying the `.spl` source. An imperative Python script that interleaves LLM calls with control flow cannot be rewritten by either kind of optimizer without understanding the programmer's intent; a declarative specification makes that intent explicit in the IR, which is what both the shipped static optimizer and the future runtime optimizer operate on. The separation of specification (`.spl`) from execution (adapter/kernel) is what makes this possible: the runtime is free to choose *how* to execute because the source only declares *what* to execute.

The same `.spl` source compiles to idiomatic code in four target frameworks via `spl3 splc` (`--target langgraph`, `go`, `typescript`, `pocketflow`). The compiler reads the AST directly and maps each construct to the target's natural idiom — `StateGraph` nodes for LangGraph, goroutines for Go parallel branches, `Promise.all` for TypeScript concurrency, and the prep/exec/post *ETL node* pattern for PocketFlow. This cross-framework compilation demonstrates that the declarative AST is a genuine intermediate representation, not a Python-specific wrapper: the same workflow specification produces native asynchronous code in each target's natural concurrency model. Full compilation examples and runtime details (kernel substrates, adapter bootstrap, template resolution, ecosystem tooling) are provided in Appendices B, F, and G.

**PocketFlow recipe migration.** PocketFlow [26] is a minimalist open-source Python framework for LLM orchestration that organizes computation as directed graphs of ETL nodes (`prep`/`exec`/`post`). SPL supports the reverse direction as well as forward compilation: existing PocketFlow Python recipes can be migrated to `.spl` via a two-phase pipeline — (1) an LLM reads the Python source and writes a natural-language functional specification, then (2) `spl3 text2spl` generates `.spl` source from that specification, using the cookbook's indexed recipe pairs as few-shot examples. Applying this pipeline to the PocketFlow open-source cookbook produced 42 migrated recipes (`cookbook-pocketflow/`), spanning patterns including supervisor agents, agentic RAG, MCP tool integration, self-healing workflows, text-to-SQL, Agent-to-Agent (A2A) protocol, and async batch processing. These recipes have not yet been individually validated against their Python originals; full validation and promotion to the main cookbook is planned as a near-term priority. The migration demonstrates a practical path to growing the SPL recipe registry organically: any sufficiently documented LLM orchestration codebase — regardless of the source framework — can be translated to `.spl` without manual rewriting.

## 6 Empirical Evaluation

### 6.1 Experiment Design

We evaluate SPL's deterministic integration using a controlled dual-arm experiment on symbolic mathematics. The experiment covers verifier rungs R1 (SymPy) and R2 (SageMath); R3 (Lean 4) is demonstrated as a design contribution with passing unit tests but is not included in this grid (see C4 in §1.3). The benchmark consists of 20 problems spanning six difficulty tiers (T0–T5), split across two symbolic backends:

| Tier | Backend | Category | Example |
|------|---------|----------|---------|
| T0 | SymPy | Single-step polynomial | $d/dx\,(x^4 - 2x^2 + 1)$ |
| T1 | SymPy | Multi-step polynomial chains | expand $\to$ factor $\to$ solve |
| T2 | SymPy | Transcendental / limits / series / trig | $\sin(x)/x$ as $x \to 0$; Taylor $\sin(x)$ deg 5 |
| T3 | Sage | Integration / linear systems / eigenvalues | $\int \sqrt{4-x^2}\,dx$; eigenvalues of $[[1,2],[3,4]]$ |
| T4 | Sage | Laplace transforms / ODEs / summation / roots | $\mathcal{L}\{e^{-2t}\}$; $y' = y$, $y(0) = 1$; $\sum 1/n^2$ |
| T5 | Sage | Expert ODE + transform verification | $y'' - 3y' + 2y = 0$;  $\mathcal{L}^{-1}\{s/(s^2+4)\}$ + verify |

T0–T2 problems (10 total) route to the SymPy kernel; T3–T5 (10 total) route to the SageMath kernel. The backend is a per-problem parameter; the `.spl` workflow is identical across both.

Ten models were evaluated: `sonnet-4-6` (Anthropic, cloud API), `gemma3` and `gemma4:e2b` (Google, local via Ollama), `qwen2.5` (Alibaba, local), `deepseek-v2:16b` (DeepSeek, local), `phi3` and `phi4` (Microsoft, local), `llama3.2` (Meta, local), `lfm2.5` (Liquid AI, local), and `rnj-1` (Essential AI [25], an 8B dense Transformer based on Gemma 3, specialized for STEM and coding, local via Ollama). Models with mandatory extended chain-of-thought (qwen3, deepseek-r1) were excluded: the thinking trace exhausts the token budget before any structured output is emitted, making them fundamentally incompatible with the `expr|op` contract (see Appendix H). Each of the ten models ran all 20 problems under two arms:

- **Solver arm** (`enable_solver=true`): the model decomposes the problem into `expr|op` steps; the symbolic kernel (SymPy or Sage) executes each step exactly; the verified chain is fed back to the model for narration. Pass = status `complete` (all steps kernel-verified end-to-end). **This pass criterion is execution-verified, not independently semantically verified**: it certifies that every decomposed step ran without error in the symbolic kernel, not that the resulting answer was checked against a known-correct value for that problem. The two are distinct — a chain can execute cleanly while still encoding an incorrect decomposition of the original problem. Independent ground-truth checking via a registered `verify(result, problem)` predicate is a separate SPL mechanism (`ASSERT`, demonstrated on the constraint-optimization and code-synthesis domains in §7.5; implementation patterns in Appendix I) and is not the pass oracle used in this experiment. **We quantify this gap directly.** A post-hoc round-trip re-verification — substituting each Pattern-1-pass result back into the original problem's defining relation, detailed as Pattern 2 in Appendix I — was run against every solver-arm row scored `pass` across both runtimes (Python and Go, §5): of 2,285 Pattern-1-pass rows, **86.7% are also round-trip verified, 9.1% fail round-trip despite passing Pattern 1** (a genuine false-positive rate on the execution-only criterion), and 4.2% are unparseable logging artifacts excluded from the count. A fair, deduped, apples-to-apples comparison using the enhanced (round-trip-instrumented) recipe on both runtimes at matched scale narrows this further to 88.3% (Python) and 88.8% (Go) verified (§5). We report this here, at first mention of the pass criterion, rather than only in Appendix I, because it is the single most load-bearing correction to the pass rates reported below.
- **LLM-only arm** (`enable_solver=false`): the model solves and narrates directly with no kernel involvement. Pass = non-empty response returned (status `complete` or `unverified_success`). This

arm measures *output production* — the model's ability to generate a complete response — not mathematical correctness. Mathematical accuracy is deliberately left unverified: the comparison isolates what SPL's deterministic mode adds (machine-verified correctness) against a baseline that represents what the LLM produces on its own.

Two sessions were run: a pilot (r=1, 400 cells) and a repeated run (r=3, 1200 cells); session identifiers are given in Appendix E, Table E.1. Results below are from the r=3 session; the pilot is retained in Appendix E for comparison. The 80-recipe cookbook, experiment runner, and raw result CSVs for both sessions are available in the project repository (see Conclusion).

## 6.2 Results

**Pass rate and latency by model and arm** (mean over 3 runs/cell, sorted by solver pass rate; 95% bootstrap CI on solver pass rate, 10k resamples):

| Model | Pass rate (%) | | | | Latency | | |
|---|---|---|---|---|---|---|---|
| | LLM-only | Solver | 95% CI | Δ | LLM-only | Solver | Δ |
| gemma4:e2b | 97 | 93 | [87, 98] | −3 | 11.9s | 14.8s | +25% |
| sonnet-4-6 | 100 | 85 | [75, 93] | −15 | 13.1s | 9.7s | −25% |
| rnj-1 | 100 | 82 | [72, 90] | −18 | 7.4s | 6.4s | −13% |
| gemma3 | 100 | 73 | [62, 83] | −27 | 4.7s | 4.6s | −1% |
| qwen2.5 | 100 | 72 | [60, 83] | −28 | 8.6s | 3.9s | −55% |
| phi4 | 100 | 67 | [55, 78] | −33 | 27.3s | 12.1s | −56% |
| llama3.2 | 100 | 65 | [53, 77] | −35 | 5.0s | 3.0s | −41% |
| deepseek-v2:16b | 100 | 53 | [40, 65] | −47 | 16.8s | 7.9s | −53% |
| lfm2.5 | 77 | 37 | [25, 50] | −40 | 7.1s | 12.2s | +71% |
| phi3 | 100 | 32 | [20, 43] | −68 | 7.4s | 4.1s | −45% |

LLM-only "pass" = non-empty response (unverified). Solver "pass" = complete symbolic chain (machine-verified by SymPy or Sage kernel). The two arms measure different things: output production vs. verified correctness; do not compare pass rates across columns as accuracy estimates. **Latency caveats:** (1) solver-arm latency includes early exits on `solver_error`, so lower latency for lower-scoring models partly reflects *failing faster*, not genuine speedup; a cleaner comparison would use passing runs only. (2) `sonnet-4-6` latency is measured via `claude_cli` (shell invocation) while local models use direct HTTP; cross-model latency comparisons are not instrument-normalized.

## 6.3 Findings

**F1 — Output-fence cleaning eliminated plan_format_error entirely.** In a preliminary run, phi4 produced its decomposition plans wrapped in markdown fences (` ```plaintext ... ``` `), causing Syntax-Errors in the kernel and 0% solver pass. Adding `strip_fences()` — a regex-based fence stripper applied immediately after `decompose_problem()` — resolved this. Across both the 400-cell pilot and the 1200-cell repeated run, zero failures were attributed to `plan_format_error` in any model. The dominant failure mode is `solver_error` (kernel-rejected expressions): *surface* format compliance (markdown fences, whitespace) is not the bottleneck — *semantic* expression correctness at the kernel boundary is. Models that fail are not failing to produce `expr|op`-shaped text; they are producing expressions the symbolic engine cannot evaluate (wrong variable names, unsupported operations, malformed syntax).

**F2 — solver_error is the primary failure mode; lfm2.5 is the plan_error outlier.** Every failure in the solver arm is `solver_error` or `plan_error` — never `plan_format_error`. Models that fail are generating syntactically valid `expr|op` plans but producing expressions the kernel cannot evaluate: wrong variable names, unsupported operations for the active backend, or malformed expression syntax (e.g., `^` instead of `**`). `lfm2.5` is the outlier: 26/60 solver failures are `plan_error` (invalid step structure), not `solver_error`, indicating a different failure mode — plan generation rather than expression evaluation.

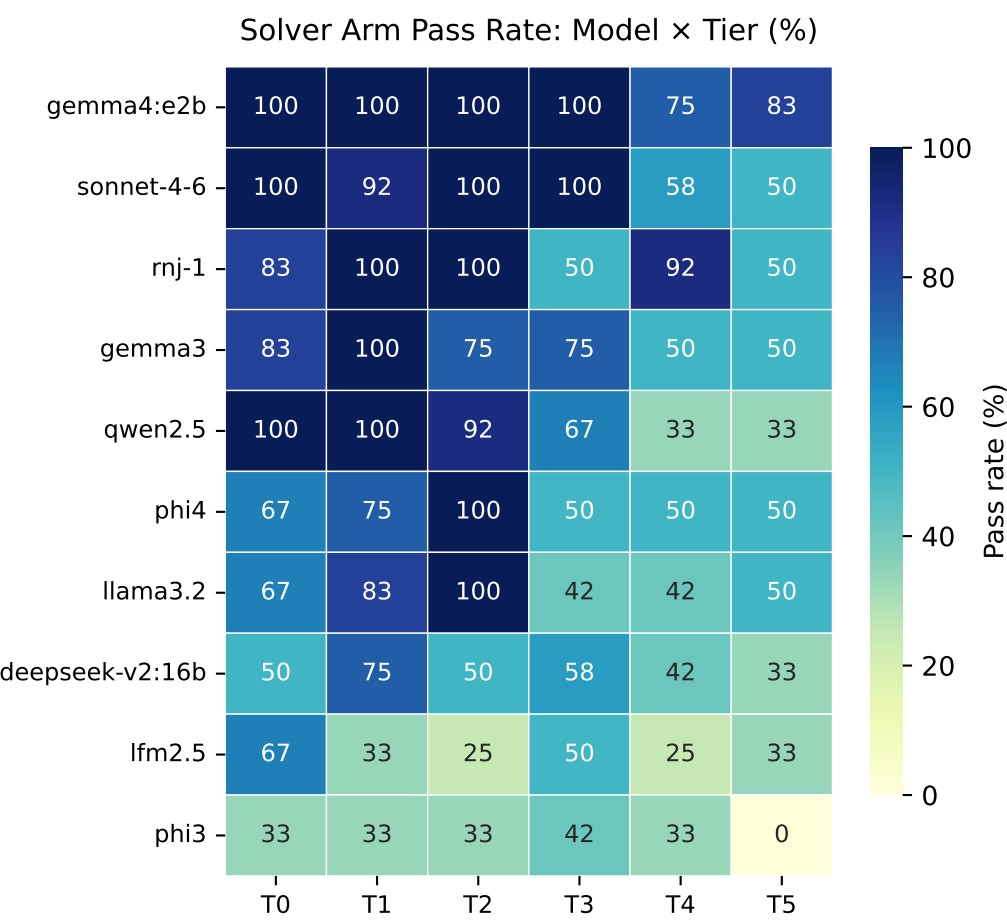

Figure 2: Solver Arm Pass Rate: Models (rows, sorted by aggregate) × tiers (columns). Diagonal gradient shows capability hierarchy; vertical gradient shows backend difficulty (SymPy 78.5% vs Sage 54.9%), though this conflates backend API familiarity with intrinsic tier difficulty (see F4).

**F3 — gemma4:e2b achieves the highest solver-arm pass rate; sonnet-4-6 is the most stable model.** `gemma4:e2b` scores 93% solver vs 97% LLM-only — the smallest gap between verified and unverified pass rates observed in this experiment. The solver-arm results are consistent with the hypothesis that symbolic delegation narrows the capability gap between small and large models; however, architecture, training data, and solver familiarity co-vary across the ten models, and a controlled study holding those factors constant is needed to confirm the mechanism. `sonnet-4-6` scores exactly 85% in both the r=1 pilot and the r=3 repeated run, showing stable results. `rnj-1` drops from 90% (pilot) to 82% (repeated), revealing pilot variance. `phi4` shows the largest pilot inflation: 85% (r=1) → 67% (r=3), an 18-point drop indicating the value of repeated runs for ranking stability. **Note on ranking confidence:** the 95% bootstrap CIs for `gemma4:e2b` [87, 98] and `sonnet-4-6` [75, 93] overlap substantially; the observed ranking difference between these two models may not be statistically significant at this sample size ($\approx$60 runs/model).

**F4 — Sage problems are structurally harder for the solver arm, with a confound.** SymPy problems (T0–T2): 78% aggregate solver pass rate (95% CI: [73, 82]). Sage problems (T3–T5): 54% aggregate solver pass rate (95% CI: [48, 60]). The Sage backend exposes harder decomposition demands: Laplace transforms, first- and second-order ODEs, infinite sums, and inverse transform verification require the model to correctly name Sage-specific operations (`laplace`, `inverse_laplace`, `solve_system`) and format expressions the Sage kernel can parse. T4 (50%) and T5 (43%) are the hardest tiers across all models. **Confound:** the SymPy/Sage gap conflates two factors — Sage is a less familiar API surface for most models *and* is assigned the intrinsically harder mathematics (T3–T5). Disentangling the two would require running overlapping problems through both backends. Per-tier numbers (T0–T5) should be read as illustrative of the difficulty gradient rather than statistically precise estimates — each tier contains 2–4 problems, yielding 6–12 runs per model-tier cell. The paper's primary contribution is the two-mode language design; the experiment demonstrates the architecture's feasibility and surfaces the format-compliance hierarchy, not an exhaustive benchmark of model capabilities.

**F5 — Solver arm accelerates most models, but early-exit confounds the comparison.** Seven models are faster in the solver arm, led by `qwen2.5` (-55%), `phi4` (-56%), `deepseek-v2:16b` (-53%). The kernel short-circuits LLM chain-of-thought: once decomposition is done, the symbolic engine resolves the chain without further LLM calls, and early-exit on `solver_error` skips the narration call entirely. However, this means part of the speedup is an artifact of *failing faster* — a `solver_error` exits before the narration `GENERATE`, producing lower mean latency for lower-scoring models. A cleaner comparison would report latency on *passing runs only*; we report all-run means here for completeness. Additionally, `sonnet-4-6` latency is measured through `claude_cli` (shell invocation) while local models use direct HTTP, so cross-model latency comparisons are not instrument-normalized. `gemma4:e2b` (+25%) and `lfm2.5` (+71%) are slower — `gemma4:e2b` because high accuracy means it completes chains and pays the full narration call; `lfm2.5` because its 26 plan_errors require longer retry attempts before failing.

The SPL status codes (`plan_error`, `solver_error`, `complete`) make per-category failure analysis automatic — each cell records its failure mode, enabling per-tier, per-backend, and per-model breakdown without log parsing.

Extended discussion of the evaluation-design choices behind these numbers — the pass/fail asymmetry between arms, why scoring LLM-only accuracy is itself hard, the human-verified subsample (E1), ranking stability across repeated runs, and the thinking-mode exclusion — is provided in Appendix I to keep this section focused on results; see §8.

# 7 Discussion

## 7.1 Completing the Computation Matrix

Table 2 situates SPL against existing systems on the computation matrix from Section 1:

| System | Probabilistic (design) | Probabilistic (runtime) | Deterministic (design) | Deterministic (runtime) |
|---|---|---|---|---|
| LangGraph | ✓ | ✓ | — | — |
| AutoGen | ✓ | ✓ | — | — |
| DSPy | ✓ | ✓ | — | — |
| AlphaGeometry | — | ✓ | ✓ | ✓ (geometry only) |
| **SPL** | ✓ | ✓ | ✓ | ✓ **(any domain)** |

No prior workflow language fills all four cells for arbitrary domains. AlphaGeometry fills all four but is domain-locked to geometry. SPL fills all four for any domain with a Python-callable verifier, with the boundary between cells declared in source rather than implicit in framework internals. The programmer writes one `.spl` file; the runtime resolves which cells are active at invocation — a difference of kind, not degree. Correctness-vs-speed trade-offs become deployment decisions, not source-embedded choices.

## 7.2 The System 1/2 Decomposition

SPL's two-mode architecture maps onto dual-process theory: *probabilistic/intuitive* (System 1, LLM) versus *deterministic/mechanical* (System 2, kernel). System 1 reads problems, chooses approaches, and narrates results; System 2 executes algebra, checks identities, and certifies proofs. The empirical results (Finding F2 and the results table in §6.2) expose a System 1/2 boundary mismatch: models with strong System 1 reasoning (100% LLM-only pass) fail at the System 1 format-mapping task SPL requires — translating problem statements into structured `expr|op` plans the kernel can evaluate. This surfaces a hierarchy: reasoning ability → format compliance → verifier access.

## 7.3 The Structured Output Bottleneck

The semantic format-mapping gap (Finding F2) extends beyond symbolic mathematics: any structured-output contract exposes a capability hierarchy where models lose verifier access if they cannot emit output the deterministic engine accepts, regardless of reasoning ability. A future feature could probe format compliance at runtime and route incapable models to the LLM-only arm; `EVALUATE` already provides the dispatch mechanism.

## 7.4 Pluggability vs. AlphaGeometry-Style Integration

AlphaGeometry [16] achieves state-of-the-art geometry theorem proving by hard-coding the neuro-symbolic loop: the neural model generates proof candidates, the symbolic engine (DD-Geometry) verifies them, and the system iterates. This tight coupling is appropriate for a single-domain system optimized for competition performance.

SPL takes the opposite design stance: the symbolic component is always an external, pluggable tool registered via `CREATE TOOL_API` or `@spl_tool`. This generality trades peak single-domain performance for domain-independence. A domain expert adds a new verifier in Python; the SPL workflow language and runtime require no modification. The same `ASSERT` semantics that check a SymPy polynomial identity can check a NetworkX graph property, a Z3 satisfiability result, or a scipy hypothesis test — any predicate the domain expert can express in Python. Empirically, open-source models like gemma4:e2b achieve 93% verified correctness on symbolic mathematics. The kernel, not the model, does the mathematical reasoning; the model only needs to reliably produce the structured `expr|op` format the deterministic engine requires. This is the design's thesis: offloading mathematical reasoning to the kernel makes *format-mapping capability*, not model scale or proprietary access, the bottleneck. When a ~2B-parameter open-source model outperforms a frontier model on verified correctness, it demonstrates that the two-mode architecture makes high-quality verified computation accessible to everyone.

### 7.5 Domain Generalization: Four Verification Oracles

The empirical evaluation in §6 focuses on symbolic mathematics (SymPy/SageMath/Lean). This section presents four additional cookbook workflows demonstrating that the `GENERATE → CALL solver → ASSERT → WHILE repair` pattern generalizes across domains wherever a machine-checkable correctness oracle exists. All four workflows are available in the project repository (see Conclusion).

**74_concept_book — Education (knowledge graph + LLM).** A single `build_concept_book.spl` workflow backed by a declarative domain graph (`{domain}_graph.yaml`) generates a complete, prerequisite-ordered textbook for any knowledge domain. The deterministic half (graph traversal, topological ordering, cycle detection via NetworkX) runs without any LLM call; the probabilistic half (prose generation per concept node) uses the LLM. Neither alone produces a coherent textbook: the LLM without the graph hallucinates prerequisite chains; the graph without the LLM produces dry node lists. The workflow has been validated on 15+ domain YAMLs (linear algebra, calculus, music theory, molecular biology, Chinese characters, and others) and generates output in any language via `-param language=zh`. It is deployed in production at `https://github.com/digital-duck/concept-book`.

**78_constraint_opt — Operations Research (PuLP/CBC solver).** An LLM reads a natural-language resource-allocation problem and generates PuLP (LP/MIP) solver code (`constraint_opt.spl`). The CBC solver executes and returns a JSON result with status `"Optimal"`, `"Infeasible"`, or `"Error"`. `ASSERT is_optimal(@solution)` gates execution: the workflow cannot reach the interpretation step unless the solver has certified the global optimum with a proof of optimality. A repair loop feeds actual solver error messages back to the LLM for code correction. The LLM never performs arithmetic; all numerical values in the output come from the solver.

**79_code_pytest — Software Engineering (pytest execution).** An LLM generates a Python module from a natural-language specification (`code_pytest.spl`). A second, independent LLM call generates pytest test cases from the *same specification* — not from the code, ensuring the tests are an independent correctness oracle. `CALL run_pytest(code, tests)` executes both in a temporary subprocess. `ASSERT all_tests_passed(@pytest_result)` gates on the actual pytest exit code. A repair loop provides the exact pytest traceback to the LLM for targeted repair. This distinguishes "code that executes" from "code that satisfies its specification" at the language level — the same distinction `ASSERT` enforces in the symbolic-mathematics experiment.

**76_lean_proof — Formal Mathematics (Lean 4 kernel).** Already part of the verifier ladder (Rung R3, §4.4). Included here to make the cross-domain pattern explicit: an LLM formalizes a prose mathematical claim in Lean 4 syntax; a second LLM call writes proof tactics; the Lean kernel checks the proof; `ASSERT` gates on the kernel verdict. A citation-path fast-pass (Mathlib lookup before proof attempt) runs first at zero token cost. This is the same `GENERATE → CALL solver → ASSERT → WHILE repair` pattern as the other three workflows, applied to the highest correctness guarantee available in the verifier ladder.

These four workflows, combined with the symbolic-mathematics experiment in §6, cover five verification domains (algebra, education, operations research, software, formal proof). In each case the ASSERT oracle is domain-appropriate and machine-checkable; the pattern requires no modifications to the SPL runtime to add a new domain — only new `CREATE TOOL_API` registrations, plus whatever domain-support code those tools need (see the Limitations paragraph on this below). Full benchmarks across all four domains are planned as a near-term priority.

### 7.6 Limitations

**Kernel startup overhead.** The IPython kernel adds ~2 seconds of cold-start latency per run. For short workflows with a single `SOLVE` step, this overhead is disproportionate. The `KernelSession` substrate (in-process `exec()`) is a lower-overhead alternative but lacks the full scientific Python environment of the Jupyter kernel.

**Verifier ladder setup cost.** R2 (SageMath) and R3 (Lean 4) require non-trivial installation beyond `pip install spl-llm`. SageMath requires a separate conda package or system install; Lean 4 requires the Elan

toolchain and Mathlib download (~2 GB). These are not barriers for a practitioner setting up a proof-carrying pipeline, but they mean the full verifier ladder is not available in a zero-configuration deployment.

**Domain-support code is not eliminated, only orchestration is.** SPL's claim is that the *workflow* — the sequencing of `GENERATE`/`EVALUATE`/`WHILE` calls against `SOLVE`/`ASSERT` checks — is expressed compactly in one declarative spec. It is not a claim that all supporting code disappears. `74_concept_book` is the clearest case: `build_concept_book.spl` (~11 KB) is the orchestration, but it is backed by several hundred lines of Python (`graph_lib.py`, `tools.py`, `style_profiles.py`) that a domain author writes once to give the workflow a graph structure and prose style to draw on — a reviewer opening that directory sees more Python than SPL by volume. `76_lean_proof` similarly depends on the framework's `spl3/lean_bridge.py` kernel bridge. We consider this domain-support infrastructure — written once per domain or per verifier, analogous to a schema or ORM layer in a SQL application — distinct from the orchestration logic SPL claims to simplify, but we do not consider that line self-evidently closed: a future `GRAPH` or `STYLE` primitive that absorbed common graph-traversal and prose-styling patterns natively would be a more complete answer than asserting the distinction and leaving the Python in place. As written, the honest claim is narrower than "no glue code": the orchestration is declarative and compact; the domain infrastructure it calls into is not, and current recipes still hand-roll it in Python.

**Pass oracle.** The r=3 results in this paper use SPL status codes as the pass oracle: a solver-arm cell passes when the symbolic kernel executes all decomposed steps without error (status `complete`). The LLM sanity gate (an independent `sonnet-4-6` judge, described in Appendix E.6) was used as an additional validation check in the 400-run pilot, not as the primary oracle. Replacing the judge with a deterministic SymPy ground-truth check of the LLM-only arm's mathematical accuracy would give a cleaner correctness comparison and is planned for the next experiment round.

## 7.7 Future Work

**Format-compliance routing.** A pre-flight probe checking whether the active model can emit the required format would gate the solver arm and eliminate format-error failures.

**Query optimizer.** SPL's declarative semantics enable a runtime optimizer: merging redundant `GENERATE` calls, caching `SELECT` results, model downgrading for cheap sub-tasks — all without modifying the `.spl` source.

**Proof-carrying workflows.** R3 (Lean 4) currently generates a proof badge but does not embed the proof artifact in the workflow's output. A proof-carrying extension would attach the Lean 4 certificate as a typed field in the `RETURN` payload, enabling downstream consumers to independently re-verify without re-running the workflow.

**Multi-domain benchmarks.** Full quantitative benchmarks for `78_constraint_opt` (operations research), `79_code_pytest` (software engineering), and `74_concept_book` (education) are planned, following the same dual-arm design used in the symbolic-mathematics experiment. Each domain has a machine-checkable oracle (LP optimality certificate, pytest exit code, graph acyclicity check) that enables the same ASSERT-based correctness measurement.

**User study.** A formal study measuring development effort (time-to-first-working-workflow, error rate, LOC) across SPL and imperative equivalents (LangGraph, PDL) would provide direct evidence for usability claims. This is an open priority.

**Distributed verifier routing.** Momagrid [22] is an open-source decentralized inference grid (GitHub repository, not peer-reviewed) that provides an SPL adapter routing LLM calls to worker nodes. As a future direction, the same routing mechanism could route `SOLVE` dispatches to verifier-specialized nodes (a node with a GPU-accelerated SageMath install, or a node with Lean 4 and Mathlib pre-warmed), enabling a distributed verifier ladder. This capability is architectural — it has not been experimentally validated in this paper.

## 8 Conclusion

Many orchestration frameworks today excel at the probabilistic side - LLM invocation, multi-agent routing, chain-of-thought scaffolding - with layers of abstraction built over decades of LLM research. The deterministic side, however, remains fragmented: symbolic computation (algebra, geometry, formal proof) lives in separate tools with no native integration path. When practitioners need both - a workflow that decomposes with an LLM and verifies the result with a solver - they resort to glue-code that is neither declarative nor portable.

SPL closes this gap through a unified declarative approach grounded in an epistemic (not temporal) reading of dual-process framing: System 1 (probabilistic, intuitive) and System 2 (deterministic, mechanical) are distinguished by their mode of computation, not their speed. A single `.spl` specification expresses both modes in the same syntax, with the same variable bindings. The runtime handles mode routing; the programmer declares *what* each step computes, not *how* or *where*. This separation enables the DODA principle: a workflow authored once runs unchanged across Ollama (local), OpenRouter (cloud), or Momagrid (distributed), with model and verifier selection deferred to invocation time.

We validate SPL through two complementary lenses. An extensive 80-recipe cookbook illustrates expressiveness across 8 workflow categories, from basic Q&A to formal proof verification, including compilation to four external orchestration targets (LangGraph, Go, TypeScript, PocketFlow). A 1200-run controlled experiment ($10 \times 20 \times 2 \times 3 = 1200$ runs; with problems spanning 6 difficulty tiers) on symbolic mathematics reveals a structural finding: the most capable models (gemma4:e2b: 93%, sonnet-4-6: 85%) reach the top of the solver arm's kernel-execution pass rate, and round-trip verification - back-substituting each passing result into the original problem's defining relation - confirms 86.7% of all Pattern-1 passes as semantically correct, isolating execution success from solved-problem correctness rather than conflating the two. Format compliance - the ability to emit structured `expr|op` plans - emerges as a separable skill from mathematical reasoning ability; models that fail format translation lose verifier access entirely, regardless of reasoning strength. The verifier ladder (SymPy for basic algebra, SageMath for advanced mathematics) makes correctness guarantees a runtime parameter, not a specification artifact; a third rung (Lean formal proof) is implemented and available via the same `ASSERT` mechanism but is not part of this empirical evaluation (§4).

A broader implication emerges: declarative composition fundamentally shifts the LLM's role from mathematical reasoner to format translator. The kernel handles derivation; the LLM handles decomposition and narration. This lowers the capability bar - a small, fast, open-source model suffices when format compliance, not reasoning depth, is the bottleneck - and enables fully offline, privacy-preserving deployment via local adapters like Ollama, where user data never leaves the device yet still benefits from machine-verified correctness. None of this is specific to mathematics: a formal grammar giving both computation modes first-class status in a single specification, and a shared variable namespace that eliminates tool-schema marshaling at the mode boundary, hold independent of the correctness-oracle questions this evaluation surfaces, and extend to other domains requiring both generative reasoning and deterministic verification - code generation and testing, scientific computation with certification, knowledge graphs with logical constraints, JSON schema validation. Source code, experimental data, and reproducibility artifacts are publicly available at `https://github.com/digital-duck/SPL.py`.

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

## Appendix A: Extended Backus-Naur Form (EBNF) Grammar

The grammar below covers the SPL language. Terminal symbols are quoted strings; non-terminals are lowercase identifiers. * = zero or more, + = one or more, ? = optional, | = alternation. The {@var} interpolation syntax (used inside string expressions for kernel dispatch) is handled at the executor level and is not shown separately.

```
(* Top-level structure *)
program         ::= (import_stmt | workflow_def | procedure_def
                    | create_fn_stmt | create_tool_stmt)*

import_stmt     ::= 'IMPORT' string_literal

workflow_def    ::= 'WORKFLOW' identifier
                    input_decl? output_decl? 'DO'
                    stmt_list
                    'END'

procedure_def   ::= 'PROCEDURE' identifier
                    input_decl? output_decl? 'DO'
                    stmt_list
                    'END'

input_decl      ::= 'INPUT' ':'? param_spec (',' param_spec)*
output_decl     ::= 'OUTPUT' ':'? param_spec (',' param_spec)*
param_spec      ::= '@'identifier type_ann? ('DEFAULT' expr)?
type_ann        ::= 'TEXT' | 'LIST' | 'SET' | 'INT' | 'FLOAT' | 'BOOL'
                    | 'IMAGE' | 'AUDIO' | 'VIDEO'

(* Statements *)
stmt_list       ::= stmt*
stmt            ::= generate_stmt
                    | evaluate_stmt
                    | while_stmt
                    | exception_stmt
                    | solve_stmt
                    | assert_stmt
                    | call_stmt
                    | call_parallel_stmt
                    | return_stmt
                    | logging_stmt
                    | assign_stmt

(* Probabilistic primitives *)
generate_stmt   ::= 'GENERATE' call_expr 'INTO' '@'identifier
                    select_clause? prompt_clause? budget_clause?

budget_clause   ::= 'WITH' 'OUTPUT' 'BUDGET' integer 'TOKENS'

select_clause   ::= 'SELECT' select_item (',' select_item)*
select_item     ::= system_role_call
                    | '@'identifier 'AS' identifier
                    | string_literal 'AS' identifier
                    | rag_call
```

```
                       | memory_call

prompt_clause    ::= 'PROMPT' string_literal

system_role_call ::= 'system_role' '(' string_literal ')'
rag_call         ::= 'rag' '(' string_literal (',' expr)? ')'
memory_call      ::= 'memory' '(' string_literal ')'

evaluate_stmt    ::= 'EVALUATE' '@'identifier
                     when_clause+ else_clause? 'END'
when_clause      ::= 'WHEN' condition 'THEN' stmt_list
else_clause      ::= 'ELSE' stmt_list
condition        ::= '=' string_literal          (* deterministic equality *)
                   | string_literal              (* semantic / LLM-judge *)
                   | 'contains' '(' string_literal ')'
                   | 'startswith' '(' string_literal ')'
                   | python_expr                 (* deterministic Python bool *)

while_stmt       ::= 'WHILE' condition 'DO' stmt_list 'END'

exception_stmt   ::= 'EXCEPTION' exception_clause+ 'END'
exception_clause ::= 'WHEN' exception_type 'THEN' stmt_list
exception_type   ::= 'HallucinationDetected' | 'ContextLengthExceeded'
                   | 'BudgetExceeded' | 'ModelRefused' | 'TimeoutError'
                   | 'WorkflowCompositionError' | identifier

(* Deterministic primitives *)
solve_stmt       ::= 'SOLVE' '@'identifier type_ann? kernel_hint? ':=' python_expr
kernel_hint      ::= 'SYMPY' | 'SAGE' | 'LEAN' | 'PYTHON'

assert_stmt      ::= 'ASSERT' python_expr
                     ('OTHERWISE' stmt_list)?

(* Workflow composition *)
call_stmt        ::= 'CALL' call_expr 'INTO' '@'identifier
call_expr        ::= identifier '(' arg_list? ')'
arg_list         ::= arg (',' arg)*
arg              ::= '@'identifier | string_literal | named_arg
named_arg        ::= identifier '=' expr

call_parallel_stmt ::= 'CALL' 'PARALLEL' call_stmt+ 'END'

(* Output and side effects *)
return_stmt      ::= 'RETURN' expr
                     ('WITH' identifier '=' expr (',' identifier '=' expr)*)?
                     (* COMMIT is a deprecated alias for RETURN *)

logging_stmt     ::= 'LOGGING' expr ('LEVEL' log_level)?
log_level        ::= 'INFO' | 'DEBUG' | 'WARNING' | 'ERROR'

assign_stmt      ::= '@'identifier ':=' expr

(* Tool and function registration *)
create_fn_stmt   ::= 'CREATE' 'FUNCTION' identifier
```

```
                          '(' param_list? ')'
                          ('RETURNS' | 'RETURN') type_ann
                          'AS' '$$' string_literal '$$'

create_tool_stmt ::= 'CREATE' 'TOOL_API' identifier
                          '(' param_list? ')'
                          ('RETURNS' | 'RETURN') type_ann
                          'AS' 'PYTHON' '$$' string_literal '$$'

param_list       ::= param_spec (',' param_spec)*

(* Expressions *)
expr             ::= '@'identifier
                   | string_literal
                   | fstring_literal
                   | number_literal
                   | bool_literal
                   | none_literal
                   | list_literal
                   | map_literal
                   | binary_expr
                   | unary_expr
                   | call_expr

fstring_literal ::= 'f"' (char | '{' '@'identifier '}')* '"'
list_literal    ::= '[' (expr (',' expr)*)? ']'
map_literal     ::= '{' (string_literal ':' expr (',' string_literal ':' expr)*)? '}'
binary_expr     ::= expr ('+'|'-'|'*'|'/'|'=='|'!='|'<'|'>'|'<='|'>='
                          |'AND'|'OR') expr
unary_expr      ::= 'NOT' expr
python_expr     ::= string_literal              (* dispatched to kernel *)

(* Lexical *)
identifier      ::= [a-zA-Z_][a-zA-Z0-9_]*
string_literal  ::= '"' [^"\\]* '"' | "'" [^'\\]* "'"
                  | "r'" [^']* "'" | 'r"' [^"]* '"'
                  | "'''" char* "'''" | '"""' char* '"""'
number_literal  ::= [0-9]+ ('.' [0-9]+)?
bool_literal    ::= 'true' | 'false' | 'True' | 'False'
none_literal    ::= 'None' | 'null'
comment         ::= ('--' | '#') char* newline
                  | '/*' char* '*/'
```

String literals use double quotes as preferred, single quotes are accepted, but can interfere with keyword highlighting in the VS Code syntax highlighter. The `{@var}` interpolation in string literals sent to SOLVE/ASSERT is resolved by the executor before kernel dispatch.

Comments are stripped before parsing. Three styles are supported: line comments with `--` or `#` (until end of line), and block comments with `/* ... */` (multi-line).

## Appendix B: Adapter System

### B.1 Mandatory Adapters

Four adapters are always available after `pip install spl-llm`. All support `--llm <adapter>:<model_id>` format. The legacy `--adapter` and `--model` flags remain available for backward compatibility.

| Adapter:Model spec | Protocol |
|---|---|
| `ollama:gemma3` | HTTP POST to `localhost:11434/v1/chat/completions` (OpenAI-compatible) |
| `claude_cli:sonnet-4-6` | Shells out to `claude` CLI; captures stdout |
| `openrouter:qwen/qwen3-8b` | HTTPS to `api.openrouter.ai`; requires `OPENROUTER_API_KEY` |
| `momagrid:gemma3` | POST `/tasks` on `MOMAGRID_HUB_URL`; poll `/tasks/{id}` for results |

`gemma` and `qwen` are used as default models throughout this paper for their continuous open-source support.

### B.2 dd-llm Bridge (Cloud Providers)

When the `dd-llm` package is installed, ten additional cloud providers are available through a unified bridge adapter. The bridge normalizes provider-specific authentication and response shapes behind the same `BaseAdapter` interface:

| Provider | Model prefix example |
|---|---|
| Anthropic | `anthropic/claude-sonnet-4-6` |
| OpenAI | `openai/gpt-4o` |
| Google | `google/gemini-2.0-flash` |
| Mistral | `mistral/mistral-large` |
| Cohere | `cohere/command-r-plus` |
| Together AI | `together/meta-llama/Llama-3-70b` |
| Groq | `groq/llama3-70b-8192` |
| Fireworks | `fireworks/accounts/fireworks/models/mixtral-8x22b` |
| Perplexity | `perplexity/sonar-large-32k-online` |
| DeepSeek | `deepseek/deepseek-chat` |

### B.3 Adapter Bootstrap Protocol

`spl3/adapters/__init__.py` resolves adapters in three-step priority order:

1. **dd-llm bridge** — if installed, registered first for all cloud model prefixes.
2. **Bespoke fallbacks** — `claude_cli` and `openrouter` registered as standalone adapters for environments without dd-llm.
3. **Always-available** — `ollama` and `momagrid` always registered last as unconditional fallbacks.

### B.4 Two-Method Interface

All adapters implement:

```
class BaseAdapter:
    def generate(
```

```
        self, prompt: str, system: str, model: str, **kw
    ) -> GenerationResult: ...

    def generate_multimodal(
        self, content: list[dict], system: str, model: str, **kw
    ) -> GenerationResult: ...
```

The executor dispatches to `generate_multimodal()` when any `INPUT` variable carries an `IMAGE`, `AUDIO`, or `VIDEO` type annotation; otherwise `generate()`. `GenerationResult` carries `content: str`, `model: str`, `latency_ms: float`, and `token_counts: dict`.

### B.5 Momagrid Response Protocol

The Momagrid hub returns a nested result object that must not be flattened:

```
{
  "state": "COMPLETE",
  "result": {
    "content": "...",
    "latency_ms": 342,
    "agent_name": "worker-node-07"
  }
}
```

`agent_name` is load-bearing for multi-node routing and audit logging.

## Appendix C: Verifier Ladder — Setup and Configuration

### C.1 Rung R1: SymPy (zero additional setup)

SymPy ships with `spl-llm`:

```
pip install spl-llm          # SymPy included
spl3 run workflow.spl --kernel --llm ollama:gemma3
```

The `--kernel` flag starts an IPython kernel with `python3` kernelspec. SymPy operations are registered as SPL tools in `cookbook/67_symbolic_math/tools.py` and are available to any workflow that imports that file or registers equivalent tools.

### C.2 Rung R2: SageMath

SageMath requires a separate install due to its size (~1 GB):

```
# Option A -- conda (recommended)
conda install -c conda-forge sagemath

# Option B -- system package (Ubuntu/Debian)
sudo apt install sagemath

# Register the SageMath Jupyter kernelspec
sage -python -m sage.repl.ipython_kernel.install

# Run with SageMath kernel
spl3 run workflow.spl --kernel --kernel-name sagemath --llm ollama:gemma3
```

SPL discovers the kernel via `ensure_kernelspec("sagemath")` in `spl3/kernel.py`. The same `SOLVE`/`ASSERT` primitives work unchanged; only the execution environment differs.

### C.3 Rung R3: Lean 4 with Lean REPL

Lean 4 and the REPL are installed via the one-shot setup script:

```
# Full install: elan + Lean 4 + leanprover-community/repl (pinned v4.30.0)
bash cookbook/tools/lean/setup_lean.sh

# Optionally with Mathlib (~2 GB download, required for library search)
bash cookbook/tools/lean/setup_lean.sh --with-mathlib
```

The script installs `elan` (Lean version manager) to `$ELAN_HOME` (default `/opt/lean`) and builds the REPL with `lake`. The SPL Lean bridge (`spl3/lean_bridge.py`) wraps the REPL in a persistent session:

```
from spl3.lean_bridge import LeanREPL

repl = LeanREPL().start()          # stdlib only
repl = LeanREPL.mathlib().start()  # with Mathlib imports
```

The bridge passes 15/15 unit tests in `tests/test_lean_bridge.py`, covering statement checking, proof verification, `exact?` citation, timeout recovery, and REPL restart.

```
# Verify installation
pytest tests/test_lean_bridge.py -v

# Run recipe 76 (the full proof pipeline)
spl3 run cookbook/76_lean_proof/lean_proof.spl --kernel --llm claude_cli
```

### C.4 Environment Variables

| Variable | Default | Purpose |
|---|---|---|
| MOMAGRID_HUB_URL | http://localhost:9000 | Momagrid hub endpoint |
| OPENROUTER_API_KEY | — | OpenRouter authentication |
| ELAN_HOME | /opt/lean | Lean toolchain root |
| SPL_LEAN_REPL_PATH | auto-detected | Path to `lean_repl` binary |
| SPL_KERNEL_TIMEOUT | 120 | Kernel execution timeout (seconds) |
| SPL_WHILE_MAX_ITER | 15 | Default WHILE loop guard |

> **Note on `SPL_WHILE_MAX_ITER`:** Complex workflows with multiple refinement loops (e.g., formalization + proof repair in Lean, or multi-step verification chains) may require higher iteration limits. If you encounter `MaxIterationsReached` exceptions, increase this value (e.g., `export SPL_WHILE_MAX_ITER=25`) to allow nested loops to complete without premature exit. The default of 15 balances safety against infinite loops with flexibility for most use cases.

## Appendix D: SPL Cookbook Recipe Catalog

The cookbook contains 80 recipes (00–79) organized across 8 categories, developed iteratively as the project evolved. Each recipe demonstrates a specific pattern or construct. Active recipes run in the batch runner (`python cookbook/run_all.py`).

**Purpose:** This recipe collection serves two roles. (1) **Feature verification**: each recipe validates a distinct SPL language construct or adapter capability, enabling rapid regression testing across new runtime versions and model providers. (2) **Code-RAG training data**: the description/source pairs are indexed and retrieved by the `text2spl` tool to guide LLM-based code generation, improving quality through in-domain examples. Community contributions are welcomed; recipes that add new constructs or clarify existing patterns strengthen both validation and the retrieval index.

**Cross-framework migration.** As a further validation of DODA, we migrated 42 recipes from the open-source PocketFlow cookbook [26] to SPL using the four-stage pipeline: (S1) `splc describe` generates a plain-English functional spec from the Python source; (S2) `text2mmd` produces a Mermaid flowchart; (S3) `mmd2spl` emits an `.spl` workflow; (S4) human review and `spl3 validate` confirm parse and semantic correctness. All 42 migrated recipes pass the automated validator (`python cookbook-pocketflow/validate_all.py`). A decentralized **workflow registry** is planned for future work, allowing teams to publish vetted recipes and share standard patterns.

### D.1 Foundations (00–09)

| ID | Name | Description |
|----|------|-------------|
| 00 | Recipe Maker | Auto-generate SPL recipes from natural language specs |
| 01 | Hello World | Basic PROMPT and WORKFLOW: single LLM call |
| 02 | Ollama Proxy | Route requests through Ollama; test local model access |
| 03 | Multilingual | Parallel LLM calls in multiple languages; vote on result |
| 04 | Model Showdown | A/B comparison of two models on identical task |
| 05 | Self-Refine | WHILE loop refinement until quality gate passes |
| 06 | ReAct Agent | Reasoning + acting: LLM thinks, then calls tools |
| 07 | Safe Generation | Content filtering and safety guardrails |
| 08 | RAG Query | Retrieval-augmented generation; query external knowledge base |
| 09 | Chain-of-Thought | Multi-step reasoning with intermediate checkpoints |

### D.2 Content Generation (10–19)

| ID | Name | Description |
|----|------|-------------|
| 10 | Batch Test | Parallel inference on a batch of prompts; collect and compare |
| 11 | Debate Arena | Two agents debate a topic; third agent judges |
| 12 | Plan and Execute | LLM decomposes task into steps; executor validates each |
| 13 | Map-Reduce | Map computation across items; reduce results with aggregation |
| 14 | Multi-Agent | Choreograph multiple specialized agents toward shared goal |
| 15 | Code Review | LLM reviews code; iterative feedback loop until approved |
| 16 | Reflection | Generate response; LLM reflects on quality; iterate if needed |
| 17 | Tree of Thought | Explore multiple reasoning paths; prune low-confidence branches |
| 18 | Guardrails | Runtime validation of LLM output against constraints |
| 19 | Memory Conversation | Multi-turn dialogue with persistent context/memory buffer |

### D.3 Analysis and Extraction (20–29)

| ID | Name | Description |
|----|------|-------------|
| 20 | Ensemble Voting | Run multiple models; majority vote on classification |
| 21 | Multi-Model Pipeline | Sequence models: model A → model B, pass-through |
| 22 | Text2SPL Demo | Natural language → SPL source code generation |
| 23 | Structured Output | Enforce JSON/XML schema on LLM response; validate format |
| 24 | Few-Shot Prompting | In-context learning: embed examples in PROMPT |
| 25 | Nested Procedures | Call workflows from within workflows; composition |
| 26 | A/B Test | Run same task on two variants; measure performance delta |
| 27 | Data Extraction | Extract structured fields from unstructured text |
| 28 | Support Triage | Classify support tickets; route to specialized handlers |
| 29 | Meeting Actions | Extract action items, attendees, dates from meeting transcripts |

### D.4 Education and Tutoring (30–39)

| ID | Name | Description |
|----|------|-------------|
| 30 | Code Generator | Generate code from natural language specification |
| 31 | Sentiment Pipeline | Analyze text sentiment; classify as positive/negative/neutral |
| 32 | Socratic Tutor | Ask clarifying questions; guide learner to answer |
| 33 | Interview Simulator | Conduct mock interview; score responses in real-time |
| 34 | Progressive Summarizer | Iteratively summarize text to target length/complexity |
| 35 | Hypothesis Tester | LLM proposes hypothesis; test against data; iterate |
| 36 | Tool-Use | Register and call custom Python functions as tools |
| 37 | Headline News Aggregator | Fetch, summarize, and rank news headlines by relevance |
| 38 | Bedrock Quickstart | AWS Bedrock adapter: example integration |
| 39 | Vertex AI Quickstart | Google Vertex AI adapter: example integration |

### D.5 Cloud and Infrastructure (40–49)

| ID | Name | Description |
|----|------|-------------|
| 40 | Azure OpenAI Quickstart | Azure OpenAI adapter: API key config and request routing |
| 41 | Human Steering | Pause for user input; incorporate feedback before proceeding |
| 42 | Knowledge Synthesis | Combine insights from multiple documents into unified summary |
| 43 | Prompt Self-Tuning | Automatically optimize prompt structure based on output quality |
| 44 | Adaptive Failover | Primary model fails → fallback to secondary model |

| ID | Name | Description |
|----|------|-------------|
| 45 | Vision to Action | Process image input; classify scene; trigger corresponding action |
| 46 | MCP Integration | Discover and invoke tools via Model Context Protocol; LLM selects tool and parameters; HTTP or local fallback |
| 47 | arXiv Morning Brief | Fetch latest papers in topic; summarize key findings |
| 48 | Credit Risk Assessment | Evaluate financial data; output risk score with explanation |
| 49 | Regulatory News Audit | Monitor regulatory news; flag relevant updates for compliance team |

### D.6 Multimodal (50–59)

| ID | Name | Description |
|----|------|-------------|
| 50 | Code Pipeline | Multimodal: parse code images $\rightarrow$ OCR $\rightarrow$ execute |
| 51 | Image Caption | Generate descriptive captions for images |
| 52 | Audio Summary | Transcribe audio; summarize transcript |
| 53 | Video Summary | Extract keyframes; caption each; synthesize summary |
| 54 | Text to Image | Generate image from text prompt (DALL-E, Stable Diffusion) |
| 55 | Text to Speech | Convert text output to audio/voice |
| 56 | Text to Video | Generate video from text script (experimental) |
| 57 | Image Format Conversion | Convert image between formats (PNG $\leftrightarrow$ JPEG $\leftrightarrow$ WebP) |
| 58 | Image Restyle | Apply style transfer or filter to image |
| 59 | Audio Format Conversion | Convert audio between codecs (MP3 $\leftrightarrow$ WAV $\leftrightarrow$ M4A) |

### D.7 Parallel and Compilation (60–69)

| ID | Name | Description |
|----|------|-------------|
| 60 | Voice Dialogue | Real-time speech $\leftrightarrow$ text; interactive conversation |
| 61 | Video to Audio | Extract audio track from video |
| 62 | Video to Image | Extract keyframes or create GIF from video |
| 63 | Parallel Code Review | Review multiple code files concurrently via CALL PARALLEL |
| 64 | Parallel News Digest | Fetch and summarize multiple articles in parallel |
| 65 | LLM Splc | Compile `.spl` to imperative code (Go/TypeScript) |
| 66 | Stock Analysis | Fetch price data; LLM generates trading signals with explanation |
| 67 | Symbolic Math | SymPy backend: algebraic problem solving (instances verified) |
| 68 | Problem Generator | Auto-generate educational problems from topic + difficulty |
| 69 | Notebook Generator | Convert `.spl` workflow to Jupyter notebook |

### D.8 Verifier Ladder and Concept-Book (70–79)

| ID | Name | Description |
|---|---|---|
| 70 | Linear Algebra Core Concepts | Build concept graph for linalg topics; trace prerequisites |
| 71 | Linear Algebra Concept-Book | Interactive HTML learning path: definitions → theorems → proofs |
| 72 | Verify arXiv References | LLM cites paper; kernel verifies bibliographic accuracy |
| 73 | Intro Geometry Concept-Book | Concept-book for geometry: points → lines → shapes → proofs |
| 74 | Generic Concept-Book | Template for concept-books on any STEM topic |
| 75 | SageMath Solver | Sage kernel: instances SymPy cannot reach (Galois, eigenvalues, etc.) |
| 76 | Lean Proof Verifier | Lean 4 + mathlib: prove statements; kernel checks proof |
| 77 | **Neurosymbolic Solver** | **Main experiment: unified workflow across SymPy → Sage → Lean rungs** |
| 78 | Constraint Optimization | LLM formulates PuLP/MIP constraints from a natural-language problem; CBC solver certifies optimality; ASSERT gates on the solver verdict |
| 79 | Code + Pytest Verification | LLM generates code and, independently, pytest tests from the same spec; ASSERT gates on the pytest exit code; repair loop on failure |

## Appendix E: Experiment — Full Results and Methodology

### E.1 Experimental Protocol

Two sessions were run on the same local workstation (Ollama for 9 local models + `claude_cli` for sonnet-4-6):

| Session | Cells | Repeats | ID |
|---|---|---|---|
| Pilot | 400 | r=1 | `exp-20260615-073849` |
| Repeated | 1200 | r=3 | `exp-20260615-191224` |

**Pass oracle:** SPL status codes — solver arm: `complete` (chain kernel-verified); LLM-only arm: `complete` or `unverified_success` (non-empty response)
**Database:** `cookbook/77_neurosymbolic/experiment_results.db` (SQLite)
**Raw execution log (r=3 session):** The full per-run SPL output log for `exp-20260615-191224` is publicly available for first-hand inspection:

```
https://raw.githubusercontent.com/digital-duck/SPL.py/refs/heads/main/cookbook/77_neurosymbolic/logs-spl/
recipe-77-log-20260615-191224.md
```

The dual-arm design uses a single `.spl` workflow (`symbolic_math.spl`) with `enable_solver` as a runtime parameter. The harness (`run_experiment.py`) iterates all (model × problem × solver_mode) cells sequentially, invoking `spl3 run` for each and persisting structured results to the DB. Status codes reported in the DB:

- `complete` — full chain verified by the symbolic kernel (solver arm) or non-empty LLM response (LLM-only arm)
- `unverified_success` — LLM-only arm produced output but with steps=0

- `solver_error` — kernel rejected one or more steps (expression evaluation failed)
- `plan_error` — plan parsed but semantically invalid (e.g., wrong step count)

### E.2 Problem Set

The 20 problems span six tiers across two backends. T0–T2 use SymPy (10 problems); T3–T5 use SageMath (10 problems). Each problem has a known ground-truth answer available in the kernel but not used as the pass oracle (status codes provide it).

| Tier | Backend | Count | Category | Examples |
|------|---------|-------|----------|----------|
| T0 | SymPy | 2 | Poly single-step | $d/dx(x^4 - 2x^2 + 1)$; simplify$(x^2 - 1)/(x - 1)$ |
| T1 | SymPy | 4 | Poly multi-step | expand$(x + 1)^2 \to$ factor; diff $3x^3 - x \to$ factor $\to$ solve |
| T2 | SymPy | 4 | Transcendental/limits/series | $\sin(x)/x \to 0$; Taylor $\sin(x)$ $n = 5$ |
| T3 | Sage | 4 | Integration/systems/eigenvalues | $\int \sqrt{4 - x^2}\, dx$; $x + y = 5, x - y = 1$; eig[[1,2],[3,4]] |
| T4 | Sage | 4 | Laplace/ODEs/sums/roots | $\mathcal{L}\{e^{-2t}\}$; $y' = y, y(0) = 1$; $\sum 1/n^2$; roots $x^4 - 1$ |
| T5 | Sage | 2 | 2nd-order ODE + verify | $y'' - 3y' + 2y = 0$; $\mathcal{L}^{-1}\{s/(s^2 + 4)\}$ verify |

### E.3 Per-Tier Pass Rates (r=3 session, `exp-20260615-191224`)

**Solver arm pass rate by model and tier (%) — mean over 3 runs/cell:**

| Model | T0 | T1 | T2 | T3 | T4 | T5 | Overall |
|-------|----|----|----|----|----|----|---------|
| gemma4:e2b | 100 | 100 | 100 | 100 | 75 | 83 | 93 |
| sonnet-4-6 | 100 | 92 | 100 | 100 | 58 | 50 | 85 |
| rnj-1 | 83 | 100 | 100 | 50 | 92 | 50 | 82 |
| gemma3 | 83 | 100 | 75 | 75 | 50 | 50 | 73 |
| qwen2.5 | 100 | 100 | 92 | 67 | 33 | 33 | 72 |
| phi4 | 67 | 75 | 100 | 50 | 50 | 50 | 67 |
| llama3.2 | 67 | 83 | 100 | 42 | 42 | 50 | 65 |
| deepseek-v2:16b | 50 | 75 | 50 | 58 | 42 | 33 | 53 |
| lfm2.5 | 67 | 33 | 25 | 50 | 25 | 33 | 37 |
| phi3 | 33 | 33 | 33 | 42 | 33 | 0 | 32 |
| **Tier avg** | **75** | **79** | **77** | **63** | **50** | **43** | — |

**LLM-only arm pass rate by model and tier (%) — mean over 3 runs/cell:**

| Model | T0 | T1 | T2 | T3 | T4 | T5 | Overall |
|-------|----|----|----|----|----|----|---------|
| sonnet-4-6 | 100 | 100 | 100 | 100 | 100 | 100 | 100 |
| rnj-1 | 100 | 100 | 100 | 100 | 100 | 100 | 100 |
| qwen2.5 | 100 | 100 | 100 | 100 | 100 | 100 | 100 |
| phi4 | 100 | 100 | 100 | 100 | 100 | 100 | 100 |
| phi3 | 100 | 100 | 100 | 100 | 100 | 100 | 100 |
| gemma3 | 100 | 100 | 100 | 100 | 100 | 100 | 100 |
| llama3.2 | 100 | 100 | 100 | 100 | 100 | 100 | 100 |
| deepseek-v2:16b | 100 | 100 | 100 | 100 | 100 | 100 | 100 |

| Model | T0 | T1 | T2 | T3 | T4 | T5 | Overall |
|---|---|---|---|---|---|---|---|
| gemma4:e2b | 100 | 100 | 100 | 92 | 92 | 100 | 97 |
| lfm2.5 | 100 | 67 | 75 | 92 | 75 | 50 | 77 |

**E.4 Failure Mode Breakdown (Solver Arm, r=3, 60 runs per model)**

| Model | complete | solver_error | plan_error | other | runs |
|---|---|---|---|---|---|
| gemma4:e2b | 56 | 0 | 4 | 0 | 60 |
| sonnet-4-6 | 51 | 8 | 0 | 1 | 60 |
| rnj-1 | 49 | 11 | 0 | 0 | 60 |
| gemma3 | 44 | 16 | 0 | 0 | 60 |
| qwen2.5 | 43 | 17 | 0 | 0 | 60 |
| phi4 | 40 | 20 | 0 | 0 | 60 |
| llama3.2 | 39 | 21 | 0 | 0 | 60 |
| deepseek-v2:16b | 32 | 28 | 0 | 0 | 60 |
| lfm2.5 | 22 | 11 | 26 | 1 | 60 |
| phi3 | 19 | 41 | 0 | 0 | 60 |

`plan_format_error` is zero across all models and both sessions. The `other` column captures rare edge cases: `sonnet-4-6` has one `unknown` status (p012/T1, run 1) and `lfm2.5` has one `narration_error` — neither is a `solver_error` or `plan_error`. `lfm2.5` is the sole `plan_error` outlier (26/60), indicating a plan-generation failure distinct from kernel expression errors. `phi3` accumulates 41 `solver_error` failures across 60 runs, the most of any model, reflecting its difficulty naming correct Sage operations.

**E.5 r=1 vs r=3 Comparison (Pilot vs Repeated Run)**

The pilot session (`exp-20260615-073849`, r=1) and repeated session (`exp-20260615-191224`, r=3) ran the identical 10-model × 20-problem × 2-arm design. Solver arm pass rates (overall %):

| Model | r=1 (pilot) | r=3 (repeated) | Δ |
|---|---|---|---|
| gemma4:e2b | 95 | 93 | -2 |
| sonnet-4-6 | 85 | 85 | 0 |
| rnj-1 | 90 | 82 | -8 |
| gemma3 | 70 | 73 | +3 |
| qwen2.5 | 75 | 72 | -3 |
| **phi4** | **85** | **67** | **-18** |
| llama3.2 | 65 | 65 | 0 |
| deepseek-v2:16b | 45 | 53 | +8 |
| lfm2.5 | 45 | 37 | -8 |
| phi3 | 30 | 32 | +2 |

Top-3 and bottom-2 rankings are stable. The largest swing is phi4 (-18 pp), revealing pilot over-estimation. `sonnet-4-6` and `llama3.2` are perfectly stable at 85% and 65% respectively, showing consistent solver behavior for those models. The r=3 session is the authoritative result used in §6.

**E.6 Judge Prompt**

The LLM sanity gate uses the following prompt for each run:

```
You are a mathematics judge. You will be given a problem and a proposed answer.
Determine whether the answer is mathematically correct.

Problem: {problem}
Proposed answer: {answer}

Reply with exactly one word: "pass" if the answer is correct, "fail" if it is not.
Do not explain. Do not add punctuation.
```

The judge model (`claude-sonnet-4-6`) is run independently of the experiment sessions and has no access to the SymPy ground truth. `qwen3` results are excluded from all tables: root-cause analysis found that `qwen3.5:9b` runs in extended thinking mode by default, exhausting its token budget on internal deliberation before emitting any structured output. This is a model-interface incompatibility, not a capability failure — the thinking trace shows correct reasoning. The experiment instead uses non-thinking alternatives (`deepseek-v2:16b`, `llama3.2`), as noted in Appendix I (§8).

## Appendix F: Compilation Pipeline — Target Examples

The `spl3 splc` compiler translates a `.spl` workflow to idiomatic code in each target framework. All examples below compile from the same source: a minimal two-mode workflow that decomposes a math problem and verifies it with SymPy.

**Source: `verify_step.spl`**

```
WORKFLOW verify_step
  INPUT: @problem TEXT
  OUTPUT: @report TEXT
DO
  GENERATE decompose(@problem) INTO @steps
    SELECT @problem AS problem
    PROMPT "Decompose into expr|op steps, one per line: {@problem}"

  SOLVE @result SYMPY := "solve_chain({@steps})"

  ASSERT verify({@result}, {@problem})
    OTHERWISE RETURN @report WITH status = 'verification_failed'

  GENERATE narrate(@result, @problem) INTO @report
    SELECT @result AS result, @problem AS problem
    PROMPT "Explain this verified result in plain language: {@result}"

  RETURN @report WITH status = 'complete'
END
```

### F.1 LangGraph Target

```
spl3 splc verify_step.spl --target langgraph

# verify_step_langgraph.py  -- generated by spl3 splc
from langgraph.graph import StateGraph, END
from typing import TypedDict

class State(TypedDict):
    problem: str
```

```
    steps: str
    result: str
    report: str
    status: str

def node_decompose(state: State) -> State:
    prompt = f"Decompose into expr|op steps, one per line: {state['problem']}"
    state["steps"] = llm_call(prompt)
    return state

def node_solve(state: State) -> State:
    state["result"] = kernel_exec(f"solve_chain({state['steps']!r})")
    return state

def node_assert(state: State) -> State:
    ok = kernel_exec(f"bool(verify({state['result']!r}, {state['problem']!r}))")
    state["status"] = "ok" if ok == "True" else "verification_failed"
    return state

def node_narrate(state: State) -> State:
    prompt = f"Explain this verified result in plain language: {state['result']}"
    state["report"] = llm_call(prompt)
    return state

def route_assert(state: State) -> str:
    return "narrate" if state["status"] == "ok" else END

graph = StateGraph(State)
graph.add_node("decompose", node_decompose)
graph.add_node("solve", node_solve)
graph.add_node("assert_gate", node_assert)
graph.add_node("narrate", node_narrate)
graph.set_entry_point("decompose")
graph.add_edge("decompose", "solve")
graph.add_edge("solve", "assert_gate")
graph.add_conditional_edges("assert_gate", route_assert)
graph.add_edge("narrate", END)
app = graph.compile()
```

**F.2 Go Target**

```
spl3 splc verify_step.spl --target go

// verify_step.go -- generated by spl3 splc
package main

import (
    "fmt"
    "splruntime"   // SPL Go runtime shim
)

type VerifyStepState struct {
    Problem string
    Steps   string
```

```
    Result   string
    Report   string
    Status   string
}

func VerifyStep(problem string) (string, error) {
    s := &VerifyStepState{Problem: problem}

    var err error
    s.Steps, err = splruntime.Generate("Decompose into expr|op steps, one per line: "+s.Problem)
    if err != nil { return "", err }

    s.Result, err = splruntime.KernelExec(fmt.Sprintf("solve_chain(%q)", s.Steps))
    if err != nil { return "", err }

    ok, err := splruntime.KernelAssert(
        fmt.Sprintf("bool(verify(%q, %q))", s.Result, s.Problem))
    if err != nil || !ok {
        return "", splruntime.CommitStatus("verification_failed")
    }

    s.Report, err = splruntime.Generate(
        "Explain this verified result in plain language: " + s.Result)
    if err != nil { return "", err }

    return s.Report, nil
}
```

**F.3 TypeScript Target**

```
spl3 splc verify_step.spl --target typescript

// verifyStep.ts -- generated by spl3 splc
import { generate, kernelExec, kernelAssert } from "./spl-runtime";

interface VerifyStepState {
  problem: string;
  steps?: string;
  result?: string;
  report?: string;
}

export async function verifyStep(problem: string): Promise<string> {
  const s: VerifyStepState = { problem };

  s.steps = await generate(
    `Decompose into expr|op steps, one per line: ${s.problem}`
  );

  s.result = await kernelExec(`solve_chain(${JSON.stringify(s.steps)})`);

  const ok = await kernelAssert(
    `bool(verify(${JSON.stringify(s.result)}, ${JSON.stringify(s.problem)}))`
  );
```

```
  if (!ok) throw new Error("verification_failed");

  s.report = await generate(
    'Explain this verified result in plain language: ${s.result}'
  );
  return s.report;
}
```

## F.4 PocketFlow Target

```
spl3 splc verify_step.spl --target pocketflow

# verify_step_pocketflow.py  -- generated by spl3 splc
from pocketflow import Flow, Node
from spl_runtime import llm_call, kernel_exec   # SPL PocketFlow runtime shim

class DecomposeNode(Node):
    # SPL: GENERATE decompose(@problem) INTO @steps
    def prep(self, shared):
        return shared["problem"]
    def exec(self, problem):
        return llm_call(
            f"Decompose into expr|op steps, one per line: {problem}")
    def post(self, shared, prep_res, exec_res):
        shared["steps"] = exec_res
        return "solve"

class SolveNode(Node):
    # SPL: SOLVE @result SYMPY := "solve_chain({@steps})"
    def prep(self, shared):
        return shared["steps"]
    def exec(self, steps):
        return kernel_exec(f"solve_chain({steps!r})")
    def post(self, shared, prep_res, exec_res):
        shared["result"] = exec_res
        return "assert"

class AssertNode(Node):
    # SPL: ASSERT verify({@result}, {@problem}) OTHERWISE RETURN ... 'verification_failed'
    def prep(self, shared):
        return shared["result"], shared["problem"]
    def exec(self, prep_res):
        result, problem = prep_res
        return kernel_exec(f"bool(verify({result!r}, {problem!r}))")
    def post(self, shared, prep_res, exec_res):
        if exec_res == "True":
            return "narrate"
        shared["status"] = "verification_failed"
        return None   # end flow early

class NarrateNode(Node):
    # SPL: GENERATE narrate(@result, @problem) INTO @report
    def prep(self, shared):
        return shared["result"]
```

```
    def exec(self, result):
        return llm_call(
            f"Explain this verified result in plain language: {result}")
    def post(self, shared, prep_res, exec_res):
        shared["report"] = exec_res
        shared["status"] = "complete"
        return None

def build_flow() -> Flow:
    decompose    = DecomposeNode()
    solve        = SolveNode()
    assert_gate  = AssertNode()
    narrate      = NarrateNode()
    decompose    - "solve"   >> solve
    solve        - "assert"  >> assert_gate
    assert_gate  - "narrate" >> narrate
    return Flow(start=decompose)
```

PocketFlow maps SPL constructs to its ETL pattern: `prep(shared)` extracts `@variables` from the shared store (Extract), `exec()` performs the LLM call or kernel dispatch (Transform), and `post()` writes results back and returns the next action string (Load). Deterministic nodes (`SOLVE`, `ASSERT`) call `kernel_exec()` in `exec()` rather than `llm_call()`; branching in `ASSERT ... OTHERWISE` is expressed as a routing return value (`"narrate"` or `None` to end early).

### F.5 DODA Invariant Across Targets

The `.spl` source is unchanged across all four targets. The compiler manages:

| SPL construct | LangGraph | Go | TypeScript | PocketFlow |
|---|---|---|---|---|
| GENERATE | StateGraph node | splruntime. Generate() | await generate() | Node.exec() → llm_call() |
| SOLVE | StateGraph node | splruntime. KernelExec() | await kernelExec() | Node.exec() → kernel_exec() |
| ASSERT ... OTHERWISE | conditional edge | if !ok branch | if (!ok) throw | post() routing: str or None |
| CALL PARALLEL | parallel nodes | goroutines + sync.WaitGroup | Promise.all([ ... ]) | BatchNode or parallel Flow |
| RETURN (COMMIT is deprecated) | END node | return / error | return / throw | None from post(); result in shared |

The DODA principle holds at the compilation level: the same declarative specification produces idiomatic, correct code in each target language without any manual adaptation.

## Appendix G: Runtime Implementation Details

### G.1 Kernel Substrates

`spl3/kernel.py` provides two execution substrates for deterministic nodes:

**IPythonKernel** runs an out-of-process Jupyter kernel via `jupyter_client.KernelManager`. It is lazy-started on the first `SOLVE` or `ASSERT` encountered during a workflow run, then held alive for the run's duration. All state — imported modules, defined variables, intermediate results — persists across steps within a session.

The kernel exposes a thread-locked `execute()` method that captures stdout and the `text/plain` repr of the last expression. Python-level errors raise `KernelExecutionError`, which the executor maps to the SPL exception hierarchy.

`KernelSession` is a lightweight in-process `exec()` substrate with a persistent namespace dict. It is used for `CREATE TOOL_API` body execution, where the overhead of an out-of-process kernel is unnecessary. Two isolation scopes are supported: `"session"` (shared namespace across the run) and `"workflow"` (fresh namespace per workflow invocation).

The kernel rung (SymPy / SageMath / Lean 4) is selected via `--kernel-name` at invocation time; setup instructions for each rung are in Appendix C.

### G.2 Template Resolution and Kernel Dispatch Harness

Before a `SOLVE` expression is sent to the kernel, the executor resolves `{@var}` interpolations:

```
def _resolve_python_template(self, expr: str, state: WorkflowState) -> str:
    return re.sub(
        r'\{@(\w+)\}',
        lambda m: str(state.get_var(m.group(1))),
        expr
    )
```

The resolved expression is then wrapped in a standard harness before dispatch:

- **SOLVE:** `_spl_solve_result = {expr}; print(str(_spl_solve_result))` — printed output is captured and assigned to the target `@var`.
- **ASSERT:** `_spl_assert_result = bool({expr}); print(_spl_assert_result)` — the executor checks the output equals `"True"`; on mismatch the `OTHERWISE` body executes.

This wrapping is invisible to the workflow author: `SOLVE` and `ASSERT` behave as if Python expressions natively return SPL variables.

### G.3 Adapter Bootstrap and Momagrid Protocol

The adapter bootstrap order in `spl3/adapters/__init__.py` is three-step:

1. **dd-llm bridge** — if installed, registered first for all cloud model prefixes (Anthropic, OpenAI, Google, Mistral, Cohere, Together, Groq, Fireworks, Perplexity, DeepSeek).
2. **Bespoke fallbacks** — `claude_cli` and `openrouter` registered as standalone adapters.
3. **Always-available** — `ollama` and `momagrid` registered unconditionally.

The Momagrid adapter submits tasks via `POST /tasks` and polls `GET /tasks/{id}`. The hub response carries a nested result object:

```
{"state": "COMPLETE", "result": {"content": "...", "latency_ms": 342, "agent_name": "worker-07"}}
```

`agent_name` is load-bearing for multi-node routing and audit logging; it must not be flattened into the top-level response.

### G.4 Ecosystem Tooling

`spl3 text2spl` converts a natural-language description to valid `.spl` source using Code-RAG: the 70+ cookbook recipes are indexed as description/source pairs and retrieved by similarity. A validation loop parses the generated output, feeds parse errors back to the LLM, and iterates until syntactically valid.

**spl3 vibe** performs one-shot NL-to-working-code generation: natural language → .spl workflow → runnable Python implementation + README + test data. The `--out-dir` flag writes all artifacts to a folder; `--adapter` selects the generating model.

Both tools are available after `pip install spl-llm` with no additional configuration.

### G.5 Lean 4 Four-Stage Protocol (SPL Source)

The full SPL implementation of the Lean rung referenced in §4, including the faithfulness-judgment stage (Stage 3 below), as shipped in `cookbook/77_neurosymbolic/symbolic_math.spl`. The faithfulness judge is not a built-in SPL primitive, only an ordinary `GENERATE` call:

```
-- Initialise the Lean REPL (once per session)
CALL run_python("from spl3.lean_bridge import LeanREPL;
                _spl_lean = LeanREPL.mathlib().start();
                print('ready')") INTO @lean_status

-- Stage 1: Formalize
GENERATE formalize_claim(@problem) INTO @lean_stmt
@check_code := f"print(_spl_lean.statement_ok(r'''{@lean_stmt}'''))"
CALL run_python(@check_code) INTO @stmt_ok

-- Stage 2: Typecheck with self-repair loop
@tries := 0
WHILE @tries < @max_tries DO
  EVALUATE @stmt_ok
    WHEN = "False" THEN
      GENERATE fix_formalization(@problem, @lean_stmt, @lean_feedback) INTO @lean_stmt
      CALL run_python(@check_code) INTO @stmt_ok
    ELSE
      @tries := @max_tries
  END
  @tries := @tries + 1
END

-- Stage 3: Judge faithfulness -- a second, independent LLM call compares
-- the formalized Lean statement against the original claim. This is a
-- probabilistic judgment, not a kernel check: an unfaithful formalization
-- can typecheck (and even prove) while still misrepresenting the claim,
-- which is exactly the class of error this stage exists to catch.
@badge := "statement_checked"
GENERATE judge_faithfulness(@problem, @lean_stmt) INTO @faithful
EVALUATE @faithful
  WHEN contains("UNFAITHFUL") THEN
    @badge := "unfaithful"
    RETURN @lean_stmt WITH status = @badge
  ELSE
    DO NOTHING
END

-- Stage 4: Prove with self-repair loop
GENERATE write_proof(@lean_stmt) INTO @lean_proof
@prove_code := f"print(_spl_lean.proof_ok(r'''{@lean_stmt}''', r'''{@lean_proof}'''))"
CALL run_python(@prove_code) INTO @proof_ok
```

```
@proof_tries := 0
WHILE @proof_tries < @max_tries DO
  EVALUATE @proof_ok
    WHEN = "False" THEN
      GENERATE fix_proof(@lean_stmt, @lean_proof, @lean_feedback) INTO @lean_proof
      CALL run_python(@prove_code) INTO @proof_ok
    ELSE
      @proof_tries := @max_tries
  END
  @proof_tries := @proof_tries + 1
END

RETURN @lean_proof WITH status = 'machine_proved'
```

The LLM's role in the Typecheck and Prove stages is proposal, not certification — correctness of the *formalized statement* and its proof is decided by the Lean kernel, not by the model. A formalization that fails to compile against Mathlib is caught at Typecheck; a claim the kernel cannot derive a proof for is caught at Prove; in either case `machine_proved` is withheld. But typechecking and provability are properties of the Lean statement in isolation — they cannot detect a formalization that is well-formed *and* provable while still misrepresenting the original natural-language claim (e.g., an inequality flipped, a quantifier's scope narrowed, a hypothesis silently dropped). That class of error is exactly what the faithfulness-judgment stage (Stage 3) exists to catch, and it is caught by a second LLM call, not by the kernel: `judge_faithfulness` is the one probabilistic gate in an otherwise kernel-certified chain. We report this plainly rather than let the badge name imply otherwise: `machine_proved` certifies that the proof is valid *for the Lean statement the faithfulness judge accepted*, not that the judge was correct to accept it.

Using a second, independent LLM call for this task is a defensible design choice — LLMs are comparatively well-suited to exhaustive, fatigue-free comparison of two texts for a mismatch, a proofreading-style task where human review is known to degrade from skimming and expectation bias — but this is a plausibility argument for the design, not a substitute for measuring it: this paper reports no independent accuracy estimate for the faithfulness judge itself, and one is needed before the claim can be strengthened further. This is what distinguishes an LLM-assisted proof pipeline from an LLM-only one: the deterministic mode's guarantee attaches to the kernel's verdict on a statement whose faithfulness is judged, not proven.

The cost/confidence trade-off across rungs is explicit: a SymPy algebraic check completes in milliseconds with zero external dependencies; a SageMath verification takes seconds and requires a separate conda environment; a Lean 4 formal proof may take tens of seconds to minutes (including Mathlib compilation on first use) and requires the Elan toolchain plus ~2 GB of Mathlib. The DODA principle lets the workflow author choose the appropriate operating point at invocation time — routine computation uses R1, publication-quality claims escalate to R3 — without modifying the `.spl` source.

### G.6 Privacy and Edge Deployment

SPL's architecture makes a fully local, zero-egress deployment a one-flag configuration choice rather than a workaround. Any local-execution stack (including plain Python calling local models) shares this property, but SPL preserves it without requiring the workflow author to write different orchestration code for the local and cloud cases. Nine of the ten models in the 1,200-run experiment are open-source and run entirely locally via Ollama or llama.cpp — no data leaves the device, no API key is required, and the SymPy/SageMath kernels are equally offline. `sonnet-4-6` is the sole cloud model and is included solely as a frontier-model reference point; it is not required for any of the 80 cookbook recipes or for the solver arm itself. A practitioner who excludes it retains a fully local, zero-egress stack.

This architecture is consistent with the data-sovereignty requirements of regulations such as FERPA and GDPR: because the `.spl` specification is backend-agnostic, switching from cloud to local execution is a one-flag change (`-adapter ollama`) with no workflow modifications. No formal privacy audit or compliance review has been conducted; we make no legal guarantee of regulatory compliance. What we do claim is a

*design stance*: SPL defaults to local, open-source execution and treats cloud API calls as optional, opt-in overrides. The 1,200-run experiment demonstrates this stance is practical at scale — 9/10 models, zero API cost, full machine-verified correctness.

## Appendix H: Model Selection for Two-Mode Workflows

Two task types arise in a declarative two-mode workflow; they suit different model behaviors.

| Property | Probabilistic | Deterministic |
|---|---|---|
| Goal | Pattern recognition, format mapping | Exact, reproducible derivation |
| Output | Approximate, context-shaped | Verifiable, same answer every run |
| Extended thinking | Liability — wastes tokens on work the kernel will do | N/A — correctness guaranteed by construction |
| Key metric | Structured output compliance | Machine-checkable correctness |
| Speed | Variable | Can be faster than probabilistic for structured problems |
| Role in SPL | LLM (GENERATE, EVALUATE) | Kernel (SOLVE, ASSERT) |

**Plan-and-Explain Pattern for Probabilistic Tasks**

The two-mode architecture enables a reusable pattern for workflows that combine LLM reasoning with deterministic verification. The **plan-and-explain** pattern decomposes the LLM's work into three stages:

1. **Plan** (probabilistic): The LLM breaks down the problem into a structured format the deterministic engine understands — e.g., decomposing a math problem into `expression|operation` steps, or translating a natural-language claim into a formal Lean statement.

2. **Execute** (deterministic): The kernel (SymPy, SageMath, Lean, or any Python-callable verifier) processes the plan step-by-step, producing exact, reproducible results. Errors are caught and reported explicitly via exception handling or status codes.

3. **Explain** (probabilistic): The LLM narrates the kernel's output in natural language, providing context, interpretation, or summary. Since the result is already verified, the LLM's role is presentation, not reasoning — it can use smaller, faster models. This stage is critical for accessibility: mathematicians and domain experts often express their work in cryptic or arcane notation (specialized symbols, terse formalism, rare terminology) that few understand. The explain step translates verified results into accessible prose, making specialized knowledge available to a broader audience — turning rare textbooks and dense proofs into comprehensible narratives without sacrificing rigor or correctness.

This pattern appears throughout the cookbook: recipe #67 (symbolic math) plans a chain of algebraic steps, verifies with SymPy, then explains the result; recipe #76 (Lean proof) formalizes a claim, checks it with the kernel, and interprets the proof badge. The pattern generalizes: any domain with a Python-callable verifier (unit testing, schema validation, graph properties, constraint satisfaction) can use plan-and-explain to combine LLM fluency with deterministic correctness.

In the SPL neurosymbolic experiment the LLM's job is **probabilistic**: recognize the problem type and map it to `expr|op` format (plan stage). The kernel's job is **deterministic**: execute each step exactly (execute stage). The LLM then explains the verified result. Thinking-mode models (qwen3.5:9b, deepseek-r1) apply deterministic-style deliberation to the plan stage, exhausting their token budget before emitting any structured output — a mismatch that plan-and-explain avoids by making the plan stage a format-translation task, not a reasoning task.

The broader implication: declarative composition lowers the LLM capability bar for math workflows. Without SPL, solving calculus requires a model that reasons deterministically. With SPL and the plan-and-explain pattern, it requires only a probabilistic translator — the kernel handles the derivation, and a smaller model handles the explanation.

## Appendix I: The `verify()` Predicate — Implementation Notes

The `verify({@result}, {@problem})` construct shown in the `verify_step.spl` listing (Appendix F, §8) and in the worked example of §3.5 is not a built-in SPL primitive. It is a **user-registered Python callable** that the kernel invokes through the `ASSERT` mechanism. This appendix clarifies how it is implemented.

### Registration

Any Python function that accepts the result and problem as arguments and returns a Boolean can serve as the pass oracle. In the SPL runtime, tool functions are registered via the `CREATE TOOL_API` declaration or by decorating them with `@spl_tool` in a companion `tools.py` module. The kernel evaluates the function and gates workflow continuation on the Boolean return value.

### Three Implementation Patterns

**Pattern 1 — Kernel execution status (used for the reported solver-arm pass rate in ğ6).** In `cookbook/77_neurosymbolic/`, the pass oracle used to compute every solver-arm number reported in ğ6 is implicit: the ASSERT expression checks the *exit status* of `CALL solve_chain()` rather than an explicit equality predicate. The tool `solve_step_with_sympy()` (and its SageMath counterpart) returns `'complete'` on successful evaluation and raises `solver_error` on failure. The effective gate is:

```
ASSERT {@solve_status} == 'complete'
  OTHERWISE RETURN @result WITH status = 'solver_error'
```

In this pattern, verification and execution are the same act: the kernel either succeeds or it does not. No separate `verify()` function is required. **This is Pattern 1 throughout, uniformly, for every SymPy/Sage solver-arm run scored pass in ğ6, on both runtimes — there is no per-tier, per-backend, or per-runtime variation in the pass criterion.** The Lean arm (R3) is a separate arm governed by the badge protocol of §4.4 and Appendix G, not this `verify()` taxonomy.

**Pattern 2 — Round-trip / back-substitution (ground-truth comparison, applied post-hoc to the T0–T5 problem set).** The direct approach to ground-truth comparison is to re-derive a canonical expected answer and diff it against the result:

```
def verify(result: str, problem: str) -> bool:
    expected = compute_expected(problem)
    return simplify(parse_expr(result)
                    - parse_expr(expected)) == 0
```

This form compares *answer forms*, and several of the T0–T5 problems have more than one valid form (an indefinite integral's free constant, an ODE general solution's arbitrarily-labeled constants, an eigenvalue list's ordering), which forces case-by-case equality logic per problem. The implementation actually used instead follows the classroom "check your work" method — **round-trip / back-substitution**: substitute the result back into the *original problem's defining relation* and confirm it holds, rather than comparing its form to a precomputed answer. Concretely: a `solve` result is plugged back into the original equation and checked to evaluate to zero; an `integrate` result is differentiated and checked to reproduce the original integrand (the free constant vanishes automatically under differentiation); an ODE's returned solution is substituted back into the ODE and checked to satisfy it identically, with the general-solution problem additionally required to retain both of its independent constants (a particular solution such as bare $e^x$ satisfies the ODE but is

not the general solution asked for). All of this is performed by SymPy itself — deterministic, with no LLM in the check.

Run post-hoc (`cookbook/77_neurosymbolic/verify_roundtrip.py`) against every SymPy/Sage solver-arm row already logged in `experiment_results.db` for these problems across both runtimes (no LLM re-run), the result is: of 2,285 Pattern-1-pass rows (3,190 solver-arm rows checked in total), **86.7% are also round-trip verified, 9.1% fail round-trip despite Pattern 1 having scored them pass, and 4.2% are unparseable** (a logging/formatting artifact excluded from the count rather than asserted as a failure). Concrete false positives recovered this way include a Laplace-transform run that transformed its own already-correct answer a second time, a first-derivative run that kept differentiating past the requested order, and a general-solution run that solved the ODE but was missing one of its two required constants. Roughly one in ten Pattern-1 passes on this problem set does not, in fact, satisfy the problem it was asked to solve. A fair, deduped comparison at matched scale using the round-trip-instrumented recipe on both runtimes (§5) narrows the verified rate to 88.3% (Python) and 88.8% (Go), confirming the gap is a property of the pass criterion, not a runtime-specific artifact. Full per-problem and per-model detail is in `roundtrip_verification_report.md` alongside the script.

**Pattern 3 — Domain predicate (general case).** For non-mathematical domains, `verify()` is any domain-appropriate Boolean check — JSON schema validation, unit-test pass/fail, graph connectivity, or compiler acceptance:

```
ASSERT json.loads({@api_response}) and schema_valid({@api_response})
  OTHERWISE RETURN @result WITH status = 'schema_violation'
```

## Multi-Approach to Verification

The three patterns above are instances of a broader point worth stating explicitly: SPL does not commit `verify()` to one fixed verification technique, because no single technique is appropriate across domains. The choice of *how* to verify is itself part of the workflow author's design space, and maps onto strategies long used outside of software — the same repertoire a student is taught for checking a worked solution:

- **Back-substitution / round trip** — substitute the result into the problem's defining relation (Pattern 2 above; §4.4's discussion of what a kernel check can and cannot catch is the same idea applied to Lean).
- **Inverse-operation check** — apply the literal inverse of the last operation performed (differentiate an integral, re-transform an inverse Laplace transform) and compare to the original input; closely related to back-substitution but not identical to it in general.
- **Special-case / limiting-case sanity check** — evaluate the result at a known-easy point, a symmetry, or a dimensional/order-of-magnitude estimate, without requiring a full inverse to exist.
- **Cross-method check** — solve the same problem by a second, genuinely different technique and require agreement; catches systematic errors that repeating the same method would not.
- **Independent second-reviewer judgment** — a second, independent evaluator (human peer review, or an LLM-as-judge call blind to the first pass's reasoning) checks the result against the original claim; this is the 79_code_pytest recipe's design (an independent LLM call generates tests from the specification, not from the code) and the Lean rung's faithfulness-judgment stage (§4.4) alike. It is a probabilistic check when the reviewer is an LLM, and its accuracy is only as good as that reviewer's, measured or not.
- **Mechanical / formal proof-checking** — a deterministic kernel (Lean's typechecker and tactic checker) verifies logical validity exhaustively; this is the strongest guarantee available, but — as §4.4 discusses — it certifies the proof of the statement it is given, not that the statement faithfully represents the original claim.

These strategies are not mutually exclusive and not uniformly available: an indefinite integral admits a clean inverse-operation check; a Laplace transform admits a clean round trip; a natural-language claim's faithfulness to its own formalization admits none of the deterministic options and is left to independent

judgment. The workflow author selects the strongest verification strategy the problem actually admits, and states which one was used — this appendix's obligation to name Pattern 1/2/3 (and, per §4.4, to name the Lean badge's faithfulness gate as a judgment rather than a kernel check) is the general form of that same discipline: a correctness claim is only as strong as the verification method backing it, and the method should be named, not left implicit in the word "verified."

### Source Code Reference

The kernel tools used in the neurosymbolic experiment are implemented in `sympolic_tools.spl` (`solve_step_with_sympy()` at line 42, `solve_step_with_sage()` at line 192):

$$\text{https:}$$
$$\text{//github.com/digital-duck/SPL.py/blob/main/cookbook/77\_neurosymbolic/sympolic\_tools.spl}$$

Neither function contains a dedicated `def verify(...)` body — the verifier IS the solver. Symbolic computation success is the proof of correctness; the `ASSERT` gate catches any deviation via the status code.

### Extended Discussion of Evaluation Design

This subsection expands on the evaluation-design choices behind the pass rates reported in §6.2–§6.3.

**Pass/fail criterion and baseline asymmetry.** The solver arm pass criterion is machine-verified: status `complete` means the symbolic kernel executed every decomposed step without error, producing a chain-verified result. The LLM-only arm pass criterion is deliberately weaker: status `complete` or `unverified_success` means the model produced a non-empty response. This asymmetry is by design — the experiment does not claim "the solver arm is more accurate than the LLM-only arm." Instead, it measures the *cost of verified correctness*: given that the LLM can produce output (near-100% in the LLM-only arm), how much does adding machine-verification via the solver reduce the pass rate? The answer — from 3% (gemma4:e2b) to 68% (phi3) — reveals the format-compliance hierarchy.

**Why scoring LLM-only accuracy is itself hard.** A natural follow-up question is: how mathematically correct are the LLM-only outputs? We attempted post-hoc verification by extracting answers from LLM narrations and comparing against the solver arm's verified results using SymPy `simplify`. The attempt yielded unreliable scores (3–28% match rate) — not because the models are necessarily wrong, but because LLM outputs are *representationally polymorphic*: the same correct derivative may appear as `4x§ - 4x` (Unicode), `4*x**3 - 4*x` (Python), `$4x^3-4x$` (LaTeX), or "four x cubed minus four x" (prose). No heuristic extraction reliably recovers a canonical symbolic form from free-text narration, and even human reviewers struggle to verify correctness across these representations without a computational tool.

This difficulty is not a limitation of the experiment — it is the central observation the paper makes. The solver arm eliminates representational ambiguity entirely: the kernel operates on canonical symbolic expressions and returns machine-checkable results. The fact that *scoring* LLM-only mathematical accuracy requires either a human expert or a symbolic verifier is itself evidence for the two-mode architecture SPL proposes.

**Human-verified accuracy subsample (E1).** To provide a direct accuracy estimate alongside the non-empty-output proxy, we manually reviewed 24 LLM-only arm responses (4 models × 6 tiers, run 1 only; entries drawn from the r=3 session, Appendix E Table E.1). Of 24 responses, 19 were fully correct, 1 was partially correct (`llama3.2` T0: correct differentiation method, arithmetic slip), and 4 were wrong (`gemma4:e2b` T3: no output produced; `llama3.2` T2–T4: incorrect procedure or no usable answer). Per-tier accuracy: T0 3/4 (+1 partial), T1 4/4, T2 3/4, T3 2/4, T4 3/4, T5 4/4. These results confirm the expected pattern: high accuracy on simpler tiers (T0–T2), degrading on T3–T5 where the solver arm achieves 100% machine-verified correctness on equivalent problems for models that pass the format-compliance gate. Critically, all four failing `llama3.2` cells returned `complete` in the automated LLM-only pass-rate table, demonstrating directly that non-empty output is not a correctness proxy. The subsample log is publicly available alongside the main experiment log at the repository URL in §6.1.

**Repeated runs and ranking stability.** The r=1 pilot (400 cells) and r=3 session (1200 cells) tell a consistent story at the top and bottom of the ranking but diverge in the middle. The top-3 ranking (gemma4:e2b → sonnet-4-6 → rnj-1) and bottom-2 (lfm2.5, phi3) are stable across both sessions. The middle tier shows pilot variance of up to 18 percentage points (phi4: 85% → 67%), indicating that r=3 is necessary for reliable ranking. Full per-run comparison is in Appendix E.5.

**Thinking-mode exclusion and CoT compatibility.** Two model families (`qwen3`, `deepseek-r1`) were excluded pre-experiment: they run mandatory extended chain-of-thought that exhausts the token budget before emitting structured output, violating the `expr|op` contract. The experiment uses non-thinking alternatives (`deepseek-v2:16b`, `llama3.2`) instead.

This exclusion reflects an architectural insight, not merely a token-budget constraint. SPL intentionally offloads mathematical reasoning to the deterministic kernel (System 2), reducing the LLM's role (System 1) to that of a *format translator* — mapping natural-language problems into strict `expr|op` syntax. Extended-thinking models are optimized for open-ended, verbose deliberation, which conflicts with the concise syntactic formatting SPL requires. The empirical results confirm this design: `gemma4:e2b`, a ~2B-parameter open-source model, achieves 93% verified correctness — not because it reasons mathematically (the kernel does that), but because it reliably produces the structured format the kernel requires. For SPL's two-mode architecture, a fast, format-compliant model is the optimal tool; extended reasoning is redundant work the kernel will perform deterministically.

SPL is not inherently incompatible with chain-of-thought reasoning — a thinking model's hidden reasoning trace could be captured as a distinct `GENERATE` step whose output is discarded before the structured `expr|op` extraction step. Models with *optional* thinking modes (e.g., `qwen3` with `--no-think`) would work with SPL as-is by disabling the thinking trace at invocation time.

## Appendix J: `EVALUATE/WHILE` versus DSPy Assertions and LMQL Constraints — A Worked Contrast

Section 2 notes that DSPy and LMQL are the closest existing systems in the declarative-LLM space but differ fundamentally from SPL's `EVALUATE/WHILE` constructs. This appendix shows the same self-refine task — iteratively improve a draft until a quality gate is satisfied — expressed in each system.

### DSPy (Stanford, 2023)

```
class SelfRefine(dspy.Module):
    def __init__(self):
        self.draft  = dspy.ChainOfThought("task -> draft")
        self.critic = dspy.ChainOfThought("task, draft -> feedback")
        self.refine = dspy.ChainOfThought("task, draft, feedback -> improved")

    def forward(self, task):
        draft = self.draft(task=task).draft
        for _ in range(3):                                # Python loop -- not DSPy
            feedback = self.critic(task=task,
                                   draft=draft).feedback
            dspy.Assert("approved" in feedback.lower(),
                        "critic must approve")
            draft = self.refine(task=task, draft=draft,
                                feedback=feedback).improved
        return draft
```

`dspy.Assert` is a *training-time optimization directive*. During `dspy.compile()`, the teleprompter uses assertion outcomes to select few-shot examples. At inference time, a failed assertion triggers a configurable

backtracking policy (retry or exception propagation), not user-defined `ELSE` logic. The iteration is Python, not DSPy: the declarative boundary ends at the module interface.

### LMQL (ETH Zurich, 2022)

```
argmax
    "Draft: [DRAFT]" where len(DRAFT) > 50
    "Feedback: [FEEDBACK]" where
        "approved" in FEEDBACK.lower() or len(FEEDBACK) > 20
from "gpt-4"
```

LMQL operates at the *token generation level*. The `where` clause enforces structural constraints (length, regex, whitelist) during beam search over the output token sequence. It cannot semantically evaluate "is this draft high quality?" — that requires an LLM judgment call, which is outside the token-constraint model. Iteration requires wrapping in a Python loop outside the LMQL query.

### SPL

```
WORKFLOW self_refine
  INPUT: @task TEXT
  OUTPUT: @draft TEXT
DO
  GENERATE draft_prompt(@task) INTO @draft
  @iter := 0
  WHILE @iter < 3 DO
    GENERATE critique_prompt(@draft, @task) INTO @feedback
    EVALUATE @feedback
      WHEN contains('[APPROVED]') THEN
        RETURN @draft WITH status = 'complete'
      ELSE
        GENERATE refine_prompt(@draft, @feedback) INTO @draft
        @iter := @iter + 1
    END
  END
  RETURN @draft WITH status = 'complete'
END
```

`EVALUATE` dispatches the condition to an LLM judge at runtime; the `WHEN` clause is a semantic pattern evaluated by the model, not a token-level constraint or compile-time hint. `WHILE` is first-class iteration with a programmer-visible exit condition. The two compose naturally: each loop body can branch to an early `RETURN`, modify variables, or continue, with the stopping criterion declared in the specification rather than embedded in Python control flow.

### Structural Differences

| Feature | DSPy / LMQL | SPL |
| --- | --- | --- |
| Quality gate | Training hint (DSPy) / token constraint (LMQL) | Runtime LLM judge (`EVALUATE`) |
| Iteration | Python `for`/`while` outside the module | First-class `WHILE` construct |
| Else-branch on failure | Backtrack policy or exception | Explicit `ELSE` block |
| Specification boundary | Module interface or single query | Full workflow including control flow |

The practical consequence: a DSPy self-refine module mixes Python control flow (the `for` loop) with DSPy calls — the iterative logic is not in the declarative specification. An LMQL query cannot express iteration within the query. An SPL workflow expresses the complete iterative refinement logic as a single declarative specification that compiles unchanged to any target framework (DODA).

