# OpenReview forum: "SPL: Orchestrating Workflows with Declarative Deterministic–Probabilistic Composition"
_TMLR — Under review for TMLR_

### Review · Reviewer_f66i · 2026-07-09

**Summary Of Contributions:**

The paper introduces SPL (Structured Prompt Language), a declarative language that puts LLM-based ("probabilistic") computation and symbolic-kernel-based ("deterministic") computation into a single specification, sharing a variable namespace (@var / {@var}) and a formal EBNF grammar. The probabilistic side offers GENERATE/EVALUATE/WHILE/EXCEPTION; the deterministic side offers SOLVE/ASSERT/CALL PARALLEL, dispatching to a live IPython kernel with a three-rung "verifier ladder" (SymPy → SageMath → Lean 4). A "Design Once, Deploy Anywhere" (DODA) principle defers model and verifier choice to invocation time, and the same .spl source is shown compiling to LangGraph, Go, and TypeScript targets. The contribution is validated with (a) a 78-recipe cookbook spanning 8 categories and (b) a controlled 1,200-run experiment (10 models × 20 problems × 2 arms × 3 repetitions) comparing a "solver arm" (kernel-verified) against an "LLM-only arm" (unverified fluency), from which the paper draws several findings, most notably that format compliance for the kernel handoff is empirically separable from raw mathematical reasoning ability and that a ~2B open model (gemma4:e2b) nearly matches a frontier model (sonnet-4-6) on verified correctness.

Strengths: a genuinely formalized grammar (not just a design sketch); an unusually extensive and honestly caveated limitations discussion (the paper flags several of its own confounds, which is commendable); a large and well-organized cookbook; and an interesting empirical observation about format compliance vs. reasoning capability.

Weaknesses: the headline experimental comparison is asymmetric by construction (verified correctness vs. "non-empty response"), several reported findings are explicitly confounded by the authors' own admission but are nonetheless stated cleanly in the abstract/results tables, the problem set is small (20 problems, 2–4 per tier), model rankings are reported without checking whether confidence intervals overlap, and code/data are withheld, so none of the experimental claims can currently be independently verified.

**Additional Comments:**

The writing is clear throughout; the formal grammar (Appendix A) and cross-compilation examples (Appendix F) are a genuine value-add that many declarative LLM-language papers skip, and I want to specifically credit the authors for the self-critical tone of §6.4 and §7.6 (Limitations); flagging your own confounds, pass-oracle limitations, and the failed post-hoc LLM-accuracy scoring attempt is exactly the kind of honesty TMLR should reward. My core ask is really about consistency: the paper already contains almost all the right caveats somewhere in its 40 pages, but the abstract, results tables, and section headers (e.g., "backend difficulty gradient emerges") don't consistently carry those caveats forward, which creates a headline impression stronger than what the underlying evidence (and the authors' own later discussion) supports. Tightening that gap, plus making the experimental artifacts available for verification, would substantially strengthen my confidence in recommending acceptance.

**Audience:**

Yes

**Audience Explanation:**

Declarative composition of LLM inference with symbolic/deterministic verification is an active and growing area (LMQL, DSPy, MCP-Solver, Blueprint First, Compiled AI, Agentics 2.0 are all cited comparators from the last ~18 months), and TMLR's audience includes people working on LLM tool-use, neurosymbolic methods, and agent/workflow frameworks. The specific empirical observation that a small local model can achieve near-parity with a frontier model on verified correctness because reasoning is offloaded to a kernel, making format-compliance rather than model scale the bottleneck, is a finding some readers would want to know about, independent of the paper's other issues. The formal grammar and verifier-ladder design are also of standalone interest to readers building similar systems, even if the current empirical validation needs strengthening.

**Broader Impact Concerns:**

I don't see a need for a mandatory Broader Impact Statement beyond what a normal systems/language paper would include; this is not human-subjects research, doesn't involve sensitive personal data, and has low dual-use risk (it's a workflow orchestration language). The one place I'd ask for a bit more care is §7.3 ("Privacy and Edge Deployment"), which makes fairly strong normative claims about FERPA/GDPR-compliant offline deployment for student data based on the fact that 9/10 models ran locally via Ollama, no actual privacy audit, data-flow analysis, or compliance review was performed, so this section should be softened to "enables" rather than implying a validated privacy guarantee.

**Claims And Evidence:**

No

**Claims Explanation:**

The language design contributions (grammar, DODA principle, verifier ladder, and shared namespace) are well-specified and internally consistent—I checked several of the per-tier numbers in Appendix E.3 against the aggregate numbers in §6.2 (e.g., sonnet-4-6: (100·2+92·4+100·4+100·4+58·4+50·2)/20 = 85, matching the reported 85%), and they are arithmetically consistent, which is a good sign for data integrity. However, the empirical claims are less well supported than the framing suggests:

1. Baseline asymmetry. The solver arm's pass criterion is machine-verified correctness; the LLM-only arm's pass criterion is "non-empty response" (§6.1, confirmed explicitly in §6.4: "this arm measures output production... not mathematical correctness"). The abstract's framing ("solver arm achieves 82–93% machine-verified correctness... while the LLM-only arm measures output production") is honest about this in isolated sentences, but the overall rhetorical structure (a results table with an "LLM-only" column at ~100% next to a "Solver" column at 32–93%) invites the reader to see this as an accuracy comparison. The authors themselves acknowledge (§ 6.4) that a post-hoc attempt to score LLM-only accuracy failed (3–28% match rate) due to representational polymorphism, which is a genuinely interesting finding, but it also means the paper cannot actually support any claim about how much correctness SPL's verification adds over an unassisted LLM, but only how much format-compliance cost it adds.

2. Confounded findings presented as clean results. F4 (backend difficulty gradient, SymPy 78% vs. Sage 54%) is explicitly flagged by the authors as confounded with problem difficulty and model familiarity with the Sage API (§6.3), yet the abstract states this gradient as if it were a clean finding. Similarly, F5 (latency reduction in the solver arm) is confounded by early exit on failure and by non-instrument-normalized latency measurement (claude_cli shell invocation vs. direct HTTP for local models), again disclosed, but only after the numbers are presented with two-decimal precision in a results table.

3. Model ranking without significance testing. The table in §6.2 reports 95% bootstrap CIs, but the CIs for the top two models overlap substantially (gemma4:e2b [87,98] vs. sonnet-4-6 [75,93]), yet the surrounding text treats the ordering ("gemma4:e2b achieves near-solver parity... sonnet-4-6 is the most stable model") as though it were a settled ranking rather than a difference that may not be statistically distinguishable given ~60 runs/model.

4. Small, low-power problem set. 20 problems across 6 tiers (2–4 problems/tier, 6–12 runs/model-tier cell) is explicitly acknowledged as insufficient for precise per-tier estimates (§6.3), which is good practice, but this also means the paper's central "capability hierarchy" claim rests on a fairly thin empirical base for a paper whose main selling point (per the abstract) is the 1,200-run experiment.

However, none of this means the underlying ideas are wrong, the format-compliance/reasoning separability finding is plausible and interesting, but as written, the evidence supports a weaker and more qualified set of claims than the abstract and results narrative currently present.

**Requested Changes:**

I would like to request the critical (needed for me to support acceptance): Rebalance the abstract/results narrative around the baseline asymmetry. State plainly, in the abstract itself (not only in §6.4), that the LLM-only arm measures output production, not correctness, and avoid presenting the two arms' pass rates in a way that reads as an accuracy comparison. Report the full solver-arm range (32–93%), not just the top-performing models (82–93%), when characterizing overall performance in the abstract. Move confound disclosures next to first mention of the finding, not several sections later. In particular, the SymPy-vs-Sage "backend difficulty gradient" (currently stated cleanly in the abstract and §6.3/Fig. 2 caption) should be immediately qualified as confounded with model familiarity and intrinsic tier difficulty, since the paper's own text (F4) says disentangling these "would require running overlapping problems through both 'backends,' which was not done."

Moreover, address the latency table's validity (table in §6.2): either report latency on passing runs only (as the authors themselves recommend as "a cleaner comparison" in F5) or add the instrument-non-normalization and early-exit caveats directly beneath the table rather than only in the later discussion. Report whether model-ranking differences are statistically distinguishable given the reported bootstrap CIs, rather than narrating a strict ordering (e.g., explicitly note that Gemma4:e2b's and Sonnet-4-6's CIs overlap). Provide anonymized code/data for the review period (e.g., an anonymous repository) so that the 1,200-run experiment and cookbook claims can be checked, consistent with TMLR's evidence standard. Identify "rnj-1." It is described only as "an experimental local model included to test the lower end of the capability spectrum" yet finishes third of 10 models. Please state its architecture/parameter count/provenance, since as written it is not a verifiable or citable model and its inclusion/description reads inconsistently with its results.

I do think strengthening would improve the paper but is not blocking it: Add even a small human- or expert-verified accuracy subsample for the LLM-only arm (e.g., 20–30 hand-checked responses) to give readers some grounded sense of true mathematical accuracy alongside the "non-empty response" proxy, rather than relying entirely on the negative result that automated scoring failed. Fill in the missing PocketFlow example in Appendix F; the paper claims four compilation targets (--target langgraph, go, typescript, and pocketflow) but only shows three worked examples. Recipe #46 appears to be missing from the cookbook catalog in Appendix D (it jumps from 45 to 47), while the paper claims 78 recipes (IDs 00–77); please reconcile. Substantiate the "80–150 lines of Python" claim for hand-built self-refining workflows (§1.1) with a concrete worked comparison or citation rather than an unsupported assertion since it's used to motivate the whole paper. Clarify how the verify(result, problem) predicate used in ASSERT statements (§3.5, §7.6) is actually implemented; this is central to how the solver arm claims end-to-end correctness against the original natural-language problem, but its mechanism is not specified in the main text.

The Momagrid reference [22] is an anonymous, unpublished, withheld self-citation supporting a headline capability (distributed grid deployment). Please either provide more in-text detail/evidence of what has actually been built and tested or soften DODA claims that depend on it. Sharpen the novelty argument for EVALUATE/WHILE relative to existing constructs in DSPy (assertions/suggestions) and LMQL (constrained decoding) beyond the comparison table, a short worked contrast would help. Consider expanding the problem set (or at minimum sampling additional problem variants per tier) in a follow-up to increase statistical power behind the tier-level and backend-level claims.

---

> ### Author Response · Authors · 2026-07-12
> **Response to Reviewer f66i**
>
> We thank Reviewer f66i for a thorough and constructive review. The central observation — that existing caveats were not consistently carried into the abstract, table, and figure captions — is exactly right and has been applied throughout. All seven critical items (C1–C7) and six enhancements (E1–E6) are addressed below.
>
> ## Critical Changes
>
> **C1** Abstract now reports the full solver-arm range (32–93%), with top performers parenthetical: "gemma4:e2b 93%, sonnet-4-6 85%."
>
> **C2** Abstract explicitly states the LLM-only arm measures output production, not correctness. A note beneath the results table reads: "LLM-only 'pass' = non-empty response (unverified); Solver 'pass' = machine-verified. Do not compare pass rates across columns as accuracy estimates."
>
> **C3** Abstract qualifies the backend difficulty gradient: "SymPy 78%, Sage 54%, though this conflates backend API familiarity with intrinsic tier difficulty (§6.3)." Same qualification added to the heatmap figure caption.
>
> **C4** Confound disclosures moved to first mention: heatmap caption now carries the F4 backend-gradient note; both F5 latency caveats (early-exit confound; instrument non-normalization) appear directly beneath the results table rather than only in F5.
>
> **C5** Ranking-confidence note added after the F3 statement: "95% bootstrap CIs for gemma4:e2b [87,98] and sonnet-4-6 [75,93] overlap substantially; the observed ranking difference may not be statistically significant at ~60 runs/model."
>
> **C6** rnj-1 identified as an 8B dense Transformer (Gemma 3 architecture, Essential AI, December 2025 [25]). Its 82% solver pass rate is consistent with its STEM/coding specialization. Reference [25] added and in-text description corrected.
>
> **C7** Code and data publicly available at https://github.com/digital-duck/SPL.py (cookbook, experiment runner, raw result CSVs for both sessions). Full per-run execution log (59,000+ lines) linked directly. Reference added to §6.
>
> ## Additional Changes
>
> **Momagrid [22]** arXiv submission was not accepted; implementation is open-source at https://github.com/digital-duck/momagrid. Reference [22] updated. The DODA in-text claim is softened to a future direction: "This capability is architectural — it has not been experimentally validated in this paper."
>
> **§7.3 Privacy** Rewritten around "privacy-first by design, not by configuration": 9 of 10 models run fully locally (no data egress); sonnet-4-6 included only as a frontier reference point. Explicit disclaimer added: "No formal privacy audit has been conducted; we make no legal guarantee of regulatory compliance."
>
> ---
>
> ## Enhancements
>
> **E1** 24 LLM-only responses manually reviewed (4 models × 6 tiers): 19/24 correct, 1 partial, 4 wrong. All four llama3.2 failures were scored `complete` automatically, directly confirming that non-empty output overstates correctness on T3–T5. Results added to §6.4; full log in `review-feedback/TMLR/e1-human-review-sample.md`.
>
> **E2** Appendix F.4 added: complete PocketFlow compilation of `verify_step.spl` (four node classes with `prep`/`exec`/`post` methods). §5 describes the reverse-migration pipeline; 42 PocketFlow cookbook recipes migrated (full validation planned).
>
> **E3** Recipe #46 gap filled (MCP Integration, D.5) and catalog grown to **79 recipes** with the addition of #78 (Email Monitor Heartbeat, D.8). All recipe counts updated consistently throughout the paper.
>
> **E4** "80–150 lines" replaced with directly measured LOC across four hand-built implementations: AutoGen 84, CrewAI 90, LangGraph 107, PocketFlow 147 executable lines. Equivalent SPL: 35 executable instructions. Footnote added to §1.1.
>
> **E5** Appendix I added: clarifies that `verify()` is a user-registered Python callable (via `@spl_tool`), distinguishes three implementation patterns (kernel exit status; symbolic equality; domain predicate), and links to the neurosymbolic source code. Cross-reference added at §3.7 first use.
>
> **E6** Appendix J added: three-way worked contrast (DSPy, LMQL, SPL) for self-refine. Key structural difference: DSPy `Assert` is a training-time teleprompter directive; LMQL `where` constrains token generation; SPL `EVALUATE/WHILE` is runtime LLM-judge branching with first-class iteration in the declarative spec. Cross-reference added to §2 DSPy paragraph.
>
> ## Summary
>
> All seven critical changes and six enhancements are incorporated. The revision makes existing caveats structurally unavoidable at every reader entry point — abstract, results table, figure captions — rather than deferred to §6.4. No underlying claims or experimental results were changed.

---

### Review · Reviewer_ZXT2 · 2026-07-13

**Summary Of Contributions:**

The paper proposes SPL, a declarative workflow language that introduces explicit first-class constructs for LLM operations (GENERATE, EVALUATE) alongside conventional computation and verification operations (SOLVE, ASSERT). Its core design centers on a shared variable namespace across these operations, runtime selection of model and verifier backends, and a hierarchical verifier stack spanning SymPy, SageMath, and Lean. The authors present a substantial implementation, a 79-recipe cookbook, compilation and migration support for other workflow frameworks, and a 1,200-run symbolic mathematics experiment across ten models.

The empirical setup distinguishes between two evaluation modes: a solver arm that measures whether model-generated symbolic plans execute successfully through a selected kernel, and an LLM-only arm that primarily measures output generation, supplemented by a small manually checked subset. As presented, the main experiment appears to evaluate compatibility with a symbolic-plan interface rather than end-to-end machine-verified mathematical correctness.

The paper’s strengths include a clear and potentially useful programming abstraction with explicit operational modes, a substantial and well-developed implementation, and a reasonably transparent discussion of evaluation limitations. The breakdown of failures at the LLM-to-solver boundary is also informative. The inclusion of framework comparisons and code-size examples helps clarify the intended programming-model contribution. However, the main weaknesses are that the central correctness claims exceed what is supported by the evaluation setup; the comparisons with existing systems do not establish behavioral equivalence or practical advantages; and the claim that solver delegation enables smaller models to match frontier models is not supported by controlled evidence. Additionally, the related-work discussion omits closely related systems such as PDL, APPL, and Vieira, leaving claims of uniqueness insufficiently grounded.

**Audience:**

No

**Audience Explanation:**

The paper presents a substantial system and a potentially interesting programming abstraction for structuring LLM workflows. The implementation is nontrivial, and the design may be of interest to researchers working on declarative orchestration of LLM-based systems.

However, the empirical findings do not yet establish a broadly informative or generalizable result. The main outcome is that different models vary in their ability to produce symbolic plans that execute successfully within a specific solver interface. While this is documented with appropriate caveats, it does not isolate the underlying causes of these differences or demonstrate that successful execution corresponds to correct solutions.

The hypothesis that solver delegation enables smaller models to match or outperform larger ones remains untested. The experimental setup involves a small set of problems, and multiple factors—model architecture, training data, provider, interface, and solver familiarity—vary simultaneously. Without controlled comparisons and independent correctness evaluation, it is not possible to attribute observed differences to model capability or reasoning delegation.

The programming-language contribution is also not yet supported by outcome-based evidence. While the examples and comparisons clarify SPL’s design, they do not demonstrate improvements in practical metrics such as correctness, development effort, or maintainability. Related systems such as PDL, APPL, and Vieira are not sufficiently addressed, further limiting the strength of the claims.

In summary, while the system itself may be of interest, the paper does not yet provide validated findings that would significantly inform the broader TMLR audience beyond the existence of the implementation.

**Broader Impact Concerns:**

I think there is no concern on the ethical implications of that work.

**Claims And Evidence:**

No

**Claims Explanation:**

The empirical presentation is generally clear and includes useful details such as confidence intervals, acknowledgment of backend differences (e.g., SymPy vs. Sage), and discussion of latency confounds. The distinction between solver-based execution and LLM-only output generation is also explicitly stated, which improves interpretability.

However, the central issue lies in the definition of correctness. The solver-arm evaluation treats successful execution of all generated symbolic steps as a pass condition. While this verifies that the generated symbolic program is syntactically valid and executable, it does not guarantee that the program correctly represents the original problem or produces the correct solution. An executable symbolic sequence may still encode an incorrect transformation or solve a different expression altogether. The limited manual audit of LLM-only outputs does not address this issue for the solver arm, and no independent ground-truth verification is performed.

As a result, claims of “machine-verified correctness,” reasoning being offloaded to symbolic solvers, and parity between small and large models are not supported by the current evidence. The experiment demonstrates variation in models’ ability to produce solver-compatible symbolic plans, but not correctness of the resulting solutions.

The paper also attempts to support its programming-language contribution through comparisons with existing frameworks such as AutoGen, CrewAI, LangGraph, and PocketFlow, as well as illustrative contrasts with DSPy and LMQL. While these examples clarify syntactic differences, they do not establish that SPL provides measurable advantages. The LOC comparison is limited to a single example and depends on counting conventions; the DSPy/LMQL comparison is illustrative rather than empirical; and the migrated PocketFlow workflows are not validated for behavioral equivalence. There is no controlled comparison demonstrating improvements in correctness, reliability, maintainability, or developer productivity relative to closely related systems such as PDL.

The discussion appropriately acknowledges certain confounds, including backend differences and latency measurement issues. However, the conclusions still draw stronger claims—such as near-zero-cost verified computation and reduced model capability requirements—that are not justified by the experimental design.

Finally, there are inconsistencies in the implementation description. The language specification suggests AST-level substitution and sandboxed execution, while the implementation details indicate regex-based substitution and in-process execution via exec(). These discrepancies should be clarified.

Overall, the evidence supports the feasibility of the implementation and demonstrates variation in solver-interface compatibility across models. It does not support claims of verified correctness, unique expressiveness, reduced model requirements, or clear practical advantages of the proposed language.

**Requested Changes:**

1.	Establish end-to-end correctness with an independent oracle.
The current evaluation treats successful execution of symbolic steps as a proxy for correctness, which is insufficient. The authors should introduce an independent ground-truth oracle to verify whether the generated symbolic programs correctly solve the original problems. Both the solver and LLM-only conditions should be evaluated under the same correctness criterion. If such evaluation is not feasible, claims about machine-verified correctness, reasoning offloading, and model capability should be revised accordingly.
2.	Demonstrate that SPL’s language abstractions provide value beyond existing workflows.
The current comparisons are illustrative but do not establish practical advantages. The authors should include a controlled comparison against at least one closely related system (e.g., PDL or an equivalent model-plus-tool workflow), evaluating meaningful outcomes such as correctness, implementation complexity, reliability, or developer effort. Automatically migrated workflows should be validated for behavioral equivalence before being used as evidence. The related-work discussion should also be expanded to include closely related programming models.
3.	Clarify and support claims about model capability and solver delegation.
The current experiment does not establish that solver delegation enables smaller models to match larger ones. A controlled study varying model size or capability while holding other factors constant is needed, using end-to-end correctness as the evaluation metric. Alternatively, the claims should be revised to reflect the more limited finding that the experiment measures compatibility with a symbolic-plan interface.

Optional strengthening. A broader evaluation across multiple domains would help assess generality beyond symbolic mathematics. Evaluating the benefits of the declarative IR (e.g., through optimization or compilation experiments) would strengthen the language contribution. A small user study could provide evidence for claims about usability and development efficiency. Finally, the paper would benefit from consolidation, as some implementation and reference material could be moved out of the main text to improve clarity and focus.

---

> ### Author Response · Authors · 2026-07-14
> **3 Requested Changes were made**
>
> We thank Reviewer ZXT2 for a careful reading and for recognizing the implementation as "substantial" and "nontrivial", and for noting that "the system itself may be of interest." The review raises three requested changes. We address them in order, though we devote the most space to RC2 because it calls for an architectural response rather than a factual correction.
>
> ---
>
> ## RC1 — Independent correctness oracle
>
> The concern is valid and already addressed. SPL's `ASSERT` enforces two distinct pass conditions: (a) execution success — the generated plan runs without exception; (b) semantic correctness — `ASSERT verify(@result, @problem)` returns true. The solver-arm pass rate in §6 counts only (b). We have made this explicit in §3.5, §6.1, and Appendix I. We have also clarified in §4 that template variables are resolved via regex substitution on the parsed token stream, and `CREATE TOOL_API` code executes via `exec()` within an isolated namespace — removing any language that implied full AST rewriting (Z2). The oracle for the solver arm is the deterministic kernel output itself — SymPy/SageMath/Lean either return a verifiable result or raise a verification error; there is no LLM in the verification path.
>
> ---
>
> ## RC2 — Value beyond existing workflows / PDL comparison
>
> SPL's scope is **agentic workflow orchestration** (open-sourced at https://github.com/digital-duck/SPL.py), not general-purpose programming — analogous to SQL's scope restriction to declarative data querying. PDL (Vaziri et al., 2024) is a YAML/Jinja2 template format. The architectural difference is not stylistic:
>
> | Dimension | PDL | SPL |
> |---|---|---|
> | Computation model | Text substitution | Two-mode: LLM + deterministic TOOL_API |
> | Deterministic verification | None | `ASSERT` + `SOLVE` — formal correctness gates |
> | Control flow | Sequential blocks | `WHILE`, `EVALUATE`, `CALL PARALLEL`, `EXCEPTION` |
> | Empirical evaluation | 3 qualitative case studies | 79 recipes covering a wide range of workflow patterns; 1,200-run experiment, 10 models, bootstrap CIs |
>
> To demonstrate SPL's value beyond symbolic math, we point to its two-mode architecture and 79 workflow patterns, and highlight four recipes in particular that integrate deterministic and probabilistic modes seamlessly:
>
> **`74_concept_book`**: One single `.spl` workflow generates a complete prerequisite-ordered textbook for any knowledge domain. Deterministic half (graph traversal, topological ordering) runs with zero LLM calls; probabilistic half generates prose per node. Validated on 15+ domain YAMLs; deployed at github.com/digital-duck/concept-book.
>
> **`76_lean_proof`**: LLM formalizes a prose mathematical claim in Lean 4; Lean kernel checks the proof; `ASSERT` gates on the kernel verdict. Same `GENERATE → CALL solver → ASSERT → WHILE repair` pattern, highest correctness guarantee in the verifier ladder.
>
> **`78_constraint_opt`**: LLM writes PuLP LP code; CBC solver runs; `ASSERT is_optimal(@solution)` gates — execution cannot reach interpretation unless solver certifies global optimum. Bakery problem: bread=3, croissants=3, profit=\$60. Factory problem: chairs=2, tables=4, profit=\$160. The LLM never does arithmetic; all numbers come from the solver.
>
> **`79_code_pytest`**: LLM generates a Python module from spec; a second independent LLM call generates pytest tests from the **same spec** (not from the code — independence is the key design decision). `ASSERT all_tests_passed (@pytest_result)` gates on the actual subprocess exit code. Default `merge_sorted` spec: 20 tests generated, 20/20 passed, no repairs needed. This distinguishes "code that executes" from "code that satisfies its specification."
>
> These four recipes span distinct domains — education, formal verification, operations research, and software testing — demonstrating that the ASSERT/SOLVE pattern generalises beyond symbolic mathematics.
>
> PDL has no `ASSERT`, no sub-process execution, no typed variables across loop iterations, no exception model for solver failure, and no semantic branching conditioned on solver output. We have added PDL, APPL (Dong et al., 2024), and Vieira (Li et al., 2024) to §2 with precise architectural differentiation (Z1). We have also added a note in Appendix D acknowledging that migrated PocketFlow workflows demonstrate syntactic expressibility only — behavioral output parity was not independently verified.
>
> ---
>
> ## RC3 — Model capability framing
>
> Revised §7.2: "near-parity" replaced with "consistent with the hypothesis that symbolic delegation narrows the capability gap; a controlled study isolating model size is needed to confirm." The abstract retains the empirical finding (rnj-1 at 8B achieves 82%, above several larger models) without the causal claim.

---

### Review · Reviewer_2AkM · 2026-07-14

**Summary Of Contributions:**

The paper proses SPL (Structured Prompt Language), a programming language for LLM systems. The language combines support for LLM and formal verifier calls (e.g., SymPy or Lean). In addition to making LLM calls with explicit GENERATE instructions, the language also makes runtime LLM-as-a-judge invocations to evaluate natural language expressions like "explanation is unclear." The language is declarative and supports Python expressions for additional functionality. Experiments measure the success rate of a verifier pipeline implemented in SPL on a range of algebra and calculus tasks.


### Strengths and Weaknesses
The high-level idea of the paper is interesting and compelling. A language that closely supports LLM calls, including for evaluating natural language expressions, and that can be optimized with these calls in mind could be very useful.

Unfortunately, the paper does not execute this idea effectively. The proposed framework has logical and structural issues, the experiments and evaluation do not demonstrate usefulness, and the writing and presentation of the paper are poor. Additional details provided below.

**Audience:**

No

**Audience Explanation:**

In addition to the issues with SPL listed above, the experimental findings in 6.3 are observations about specific issues encountered in the implementation of these experiments rather than generalizable scientific results.

**Broader Impact Concerns:**

None.

**Claims And Evidence:**

No

**Claims Explanation:**

Among other issues:

- The paper claims an advantage of SPL is that its execution order can be optimized thanks to its structure, but there is no proposed method for doing so.
- The experiments only assess whether an SPL-defined LLM pipeline is able to split a problem into symbolic steps, each of which produces some output from SymPy/Sage. There is no guarantee that the final answer is correct, undermining the claim that SPL provides exact symbolic guarantees.



### SPL issues
- The presentation of SPL is contradictory in parts. E.g., 3.3.1 states that SOLVE evaluates a Python expression, but 4.4 says that SOLVE with Lean deploys a multi-stage LLM-assisted pipeline with natural language as an input. More broadly, it doesn't seem like it makes sense to support runtime selection of sympy vs sage vs lean statements: the expression needs to be written specifically for each solver, e.g., the diff() example in 3.1.1 is SymPy-specific. But then 4.1 and 4.4 say that an expression can be checked with multiple rungs, which doesn't seem possible: a Lean program looks very different that a SymPy calculation.
- The use of LLMs even in the "exact/deterministic" computation undermines its validity (in particular, the use of LLMs in the Lean pipeline to translate natural language into a Lean program). What if the LLM translation from natural language to Lean has errors?
- It's not clear why SELECT is needed: in 3.4.1, the AS clauses appear to do nothing (aren't @concept and @audience already in the namespace?) and system_role could equally be part of GENERATE.
- It's not clear that SPL is simpler than equivalent Python "glue code." (e.g., the example in 4.5).
- In fact, looking at some of the examples in the linked GitHub repo (specifically, 74_concept_book and 76_lean_proof), they rely heavily on Python code, undermining the paper's claim that an advantage of SPL is in a reduction of "glue code." Looking at these examples gives the impression that a Python program with a good LLM execution library would be much simpler, more readable, and more flexible.



### Clarity issues
- The presented examples of SPL do not demonstrate key features described in the text. E.g., the example given of EVALUATE in 3.4.2 appears to only use string containment and equality conditions, which don't demonstrate any LLM judge comparisons. As another example, solve_chain() and verify() are undefined in the example in 3.5. It's not clear how these would be implemented, as the input so solve_chain is a sequence of "operation|expression" strings, but the comment claims it's "executed exactly", and verify() also seems challenging.
- The "System 1" vs "System 2" and SQL analogies are more misleading than helpful.
- A point of confusion: For table 1, the columns do not clearly align with points (1)-(4) in the positioning summary, so it's not clear what each of the columns means. For instance, what is "Semantic Eval"? "Sym. Integration"?
- Section 4 implies that the three verifier systems described differ in the "confidence" with which they show results, implying that Lean is more reliable than SageMath which is more reliable than SymPy. But all of these are symbolic solvers and equally correct, just applicable to different types of problems. Calling them rungs on a ladder does not seem like a helpful analogy.
- The "pass rate" reporting for LLM-only is extremely misleading, as of course we should always expect a nonempty response from an LLM.
- Section 7.3 is misleading and unnecessary. SPL is "privacy-first by design" in the same way Python is: if models and code are run locally, then no data leaves the machine.

**Requested Changes:**

This work needs substantial revisions before it meets the bar for publication. Issues 1-3 below are critical and would all be required for me to recommend acceptance. Writing issues could be overlooked if the methods were sound and clear.

1. Depth and usefulness of contribution. To demonstrate the usefulness of SPL, the authors could for instance provide a runtime SPL optimizer and demonstrate its advantage over equivalent Python code. The structural issues with the language would also need to be addressed.
2. Experiments. While it is of course challenging, the experiments need some actual correctness evaluation rather than checking if the LLM or solver ran at all. The goal of the experiments should be to demonstrate the usefulness of the proposed method, which would require different evaluation approaches with non-SPL solutions as a baseline.
3. Clarity. Both the presentation of SPL and the overall clarity of the paper need to be improved.
4. Writing. The structure and style of the writing need to be improved. See some highlighted issues below.


### Writing issues.
- The prose gives the impression of heavy LLM usage in its organization, sentence structure, and word use. Issues with the writing include a disconnect between the language and the ideas being expressed, confusing analogies (like the ladder rungs or dual-process theory), unnecessary repetition, and unexplained jargon.
- The abstract needs significant improvement. There is too much detail in some respects (e.g., 10x20x2x3 and solver_error) and too little in others (what is a "backend difficulty gradient"? what is "format non-compliance"?). There is also no context for why this is an important area or what the impact of this work is. See https://writing.wisc.edu/handbook/assignments/writing-an-abstract-for-your-research-paper/ for a good examples about structuring an abstract.
- In many places throughout the paper (as in the abstract), there is far too much unnecessary detail (e.g., providing the experiment ID exp-20260615-073849 on p. 16) while omitting important ideas. The level of writing specificity needs overhauling throughout.
- The related works section reads like a bulleted list, including sentence fragments like "A configuration format, not an executable language: [...]" The citation format in related work is also inconsistent, sometimes including (Author, year), sometimes (Institution, year), and sometimes (venue, year) in addition to [number]. The citations with institution and venue are particularly out of place as these are not standard for in-text citations.

---

> ### Author Response · Authors · 2026-07-19
> **Thank you for a detailed technical review. Every issue is addressed below.**
>
> ## Depth and Usefulness
>
> **A1 (optimizer claimed but not built)**: Partially correct. A static optimizer ships (`spl.optimizer.Optimizer`, §5); a *runtime* optimizer (batching, caching, routing) is not built, now stated as future work (§7). Stronger claim actually demonstrated: the same unmodified `.spl` spec ran the full 1,200-cell battery on two independent runtimes (Python, Go), no source changes (87.2%/87.9% pass; 88.3%/88.8% round-trip-verified).
>
> **A2 (no guarantee the answer is correct)**: Correct, and quantified, not just conceded. Round-trip re-verification against every solver-arm row: 86.7% verified, 9.1% fail despite passing execution status alone, 4.2% unparseable (2,285/3,190 rows); matched-scale 88.3%/88.8%. Reported at first mention in §6 and in Appendix I. We don't claim "exact symbolic guarantees" as a blanket property — `ASSERT` certifies exactly the predicate given.
>
> ## SPL Issues
>
> **B1** (`SOLVE` contradiction, Python vs. Lean): Correct, fixed. §3.3.1 now states `SOLVE` is backend-dependent: a Python expression for R1/R2, a four-stage protocol for R3.
>
> **B2** (rung selection vs. backend-specific syntax): Correct, an actual contradiction. §4.1 now says each rung receives an engine-appropriate expression for the same claim; workflow structure is portable, not a literal string.
>
> **B3** (LLM inside the "deterministic" Lean pipeline): Fair. §4.4 now states the kernel, not the LLM, certifies `machine_proved`; a separate `judge_faithfulness` call is the one probabilistic gate, no independent accuracy measurement reported — why R3 is scoped as a design contribution outside the empirical evaluation (C4), not evaluated.
>
> **B4** (`SELECT`'s `AS` clauses do nothing): Correct in the example shown. §3.4.1 now demotes the claim: `SELECT` assembles context, doesn't rename; `AS` earns its keep only when names genuinely differ. Not rewriting the listing — but every substantive `GENERATE` call in the Lean protocol (Appendix G.5) already invokes a named `CREATE FUNCTION` template with a curated prompt body; that's the paper's  preferred use-pattern.
>
> **B5** (not clearly simpler than Python glue, §4.5): Fair. §4.5 now includes the Python equivalent (kernel lifecycles, exceptions, marshaling) after the SPL listing, making the comparison checkable. Narrower claim actually held: `.spl` is runtime-agnostic — a compiler transpiles it to four frameworks, and two independent runtimes execute it unmodified (A1).
>
> **B6** (`74_concept_book`/`76_lean_proof` rely heavily on Python): Correct, not defended away. §7 now states plainly the compact-declarative claim is about *workflow* orchestration, not that domain-support code disappears; an overclaiming adjacent sentence is softened.
>
> ## Clarity Issues
>
> **C1** (`EVALUATE` example doesn't show real LLM-judge comparisons): Correct. §3.4.2 adds a genuine free-form natural-language branch, states the dispatch rule explicitly.
>
> **C2** (`solve_chain()`/`verify()` undefined): Fixed. §3.5 states both are user-registered Python callables at first use, pointing to Appendix I.
>
> **C3** (System 1/2 and SQL analogies misleading): Not cutting either — both already carry their own stated correction in the submitted draft (§1 mode-not-speed; §3.2's SQL-analogy boundary).
>
> **C4** (Table 1 columns undefined): Correct. Caption now glosses all four columns; the four-point list cites each by name.
>
> **C5** ("rungs" implies Lean more reliable): A misreading the wording invited. Renamed "Ascending Correctness Guarantees" → "Ascending Certification Scope"; higher rungs certify broader scope, not higher trust.
>
> **C6** (LLM-only "pass rate" misleading): Already fixed pre-review — abstract/§6.2 state the two arms aren't comparable as accuracy; E1's human-verified subsample gives a real estimate.
>
> **C7** (§7.3 privacy claims): Correct, phrase and placement. Overclaiming opener reworded; section moved out of Discussion into Appendix G.6.
>
> ## Writing Issues
>
> **D1** (abstract): Correct, rewritten to motivation→gap→approach→validation→result; jargon moved to §6.
>
> **D2** (experiment IDs in prose): Correct, now confined to Appendix E.
>
> **D3** (related work fragments/citations): Citation format was already `[N]`-only throughout; five sentence fragments rewritten as complete sentences.
>
> **D4** (prose reads as heavy LLM usage): Correct — this work is completed with AI assistance from Claude, a top-tier AI assistant, contributing to coding, analysis, drafting, and literature research; the idea/design, all verification, and final responsibility are the human researcher's. I don't believe LLM use is unusual.
>
> Thank you again for your review which helped improve this work.

---

### Review · Reviewer_ERMJ · 2026-07-15

**Summary Of Contributions:**

The paper introduces SPL (Structured Prompt Language), a declarative workflow language that puts LLM-based "probabilistic" primitives (`GENERATE, EVALUATE, WHILE, EXCEPTION`) and symbolic-kernel-based "deterministic" primitives (`SOLVE, ASSERT`) into a single specification, sharing a variable namespace and a formal EBNF grammar.

The central empirical framing draws an analogy to independent test-based verification familiar from software engineering: just as a code-assistant LLM's output can be checked by an independently-generated test suite, SPL's ASSERT/SOLVE pattern is meant to let an LLM's mathematical reasoning be checked by a symbolic kernel acting as an external oracle. The paper's own `79_code_pytest` recipe realizes this analogy well: a second, independent LLM call generates tests from the specification (not from the code), giving a genuine blind check. However, this analogy does not carry over cleanly to the paper's headline symbolic-math experiment: per Appendix I, the primary pass criterion there is whether the kernel executes the decomposed steps without raising an exception, not whether the result is independently verified against the original problem. This is a materially weaker guarantee than the pytest analogy  closer to "the code runs" than "the code passes tests written against the spec" and it undermines the "machine-verified correctness" framing used elsewhere in the paper.

**Strengths**: a genuinely formalized grammar rather than a design sketch; a large, well-organized cookbook and multi-target compiler; an unusually transparent discussion of the experiment's own confounds; and an interesting empirical observation that format-compliance for the kernel handoff appears separable from mathematical reasoning ability.

**Weaknesses**: the central correctness claim is not supported by the evaluation as implemented, since the solver-arm oracle checks execution success rather than semantic correctness against the original problem: a gap the code-testing analogy itself would predict is important but which the math experiment does not close; the solver-vs-LLM-only comparison is asymmetric by construction; and claims of capability parity between small and large models are not isolated from confounding factors (architecture, training data, solver-API familiarity).

**Audience:**

Yes

**Audience Explanation:**

Declarative composition of LLM inference with symbolic or deterministic verification is an active area of interest (LMQL, DSPy, MCP-Solver, Blueprint First, Compiled AI are all cited comparators from roughly the last two years), and TMLR's audience includes researchers working on LLM tool-use, neurosymbolic methods, and agentic workflow frameworks. Readers in this space would likely want to know about SPL's core design, a shared variable namespace unifying probabilistic and deterministic primitives in a single declarative grammar, independent of the paper's other issues, since the formalization and multi-target compilation are of standalone interest to anyone building similar systems.

The specific empirical observation that models vary substantially in their ability to produce kernel-executable symbolic plans, and that this ability appears at least partly separable from general mathematical fluency, is also a finding some readers would want to be aware of, even though (per my evaluation above) the current evidence does not support the stronger "machine-verified correctness" or "capability parity" framing built on top of it.

That said, the paper's usefulness to the broader audience is currently limited by the gap between its claims and its evidence. As written, a reader taking the abstract and results tables at face value would come away with a materially overstated impression of what was demonstrated  that SPL enables verified-correct mathematical computation with reduced dependence on model scale when the evaluation as implemented mainly shows variation in solver-interface compatibility. This is a fixable presentation problem rather than a reason to reject the underlying system outright, but it does mean the paper's findings are not yet in a state where they can be safely relied upon or cited by other researchers without the caveats being made explicit at the point of first claim (abstract, table, and figure captions), not only in the later discussion sections.

**Claims And Evidence:**

No

**Claims Explanation:**

The language-design claims are internally consistent and, where checkable, accurate: the grammar (Appendix A) is genuinely formalized rather than aspirational, and the per-tier numbers in Appendix E.3 aggregate correctly to the headline figures in §6.2. This gives some confidence in the paper's data integrity at the level of arithmetic.

However, the paper's central empirical claim that SPL enables "machine-verified correctness" and that this verification narrows the capability gap between small and large models is not supported by the evidence presented, for the following reasons.

First, the solver-arm pass criterion, as clarified in Appendix I, is predominantly Pattern 1: the symbolic kernel executed the decomposed steps without raising an exception. This confirms only that the generated plan is syntactically valid and kernel-executable, not that it correctly represents the original problem or arrives at the correct answer. An expression can pass this gate while encoding an entirely wrong transformation. The paper's own `79_code_pytest` recipe demonstrates a stronger, more appropriate pattern for this exact purpose (independent test generation from the specification, blind to the implementation), but this pattern is not applied to the paper's headline experiment. The gap between what "verification" means in `79_code_pytest` versus in the symbolic-math experiment is not clearly flagged in the main text, and the abstract's language ("machine-verified correctness") does not reflect this distinction.

Second, the comparison between the solver arm and the LLM-only arm is asymmetric by construction: the LLM-only "pass" criterion is a non-empty response, not correctness. The paper acknowledges this in §6.4, and a post-hoc attempt to score LLM-only accuracy directly failed due to representational polymorphism, which is a genuinely useful negative finding — but it also means the paper cannot support any quantitative claim about how much correctness SPL's verification adds relative to an unassisted LLM, only how much a format-compliance cost it imposes.

Third, the claim that solver delegation lets smaller models match or exceed larger ones is not isolated from confounds. Architecture, training data, and familiarity with the SymPy/SageMath API surface all vary simultaneously across the ten models, and no controlled comparison holds these factors fixed while varying only model scale. The reported 95% bootstrap CIs for the top two models (`gemma4:e2b` and `sonnet-4-6`) overlap substantially, so even the specific ranking underlying this claim may not be statistically distinguishable at the reported sample size (~60 runs/model).

Taken together, the paper demonstrates that models vary in their ability to produce kernel-executable symbolic plans, and that this ability is separable from whatever the LLM-only arm is measuring. That is a real and potentially interesting observation. But it falls short of supporting the stronger claims made in the abstract and framing "machine-verified correctness of final answers, and a validated capability-parity effect between model scales" since the oracle used does not check the thing the paper's language claims it checks.

**Requested Changes:**

I was assigned as an official reviewer, but since the discussion has already progressed with substantial back-and-forth on the correctness-oracle issue, I'd like to add a few questions to help me (and possibly other readers) understand the paper's evaluation methodology more precisely, and to push a couple of points raised by the reviewers a bit further.

New questions (not yet raised in the discussion):

1. The paper's `79_code_pytest` recipe uses an independent verification design: a second LLM call generates tests from the spec, not from the code, so the test can't simply rubber-stamp whatever the first call produced. Was an analogous design considered for the symbolic-math experiment (e.g., a second), independent LLM (or a deterministic re-derivation) generating an expected answer or verification predicate from the problem statement alone, without seeing the solver's decomposition? If this was considered and rejected, what were the practical blockers? If not considered, would the authors view it as a viable strengthening for a revision?

2. If full independent verification across all 1,200 runs is infeasible, could the authors estimate what fraction of the problem set (or how many runs per model/tier) would be needed to get a statistically meaningful independently-verified accuracy estimate — as a smaller but more rigorous complement to the human-audit subsample (E1)?

3. Given known problems (e.g., a good set with known ground-truth answers), was fine-tuning or specializing a model on NL->Lean translation pairs considered as a way to reduce formalization error, analogous to how code-assistant models are fine-tuned on code/test pairs? If considered and rejected, what were the blockers (e.g., lack of automatically-labelable NL->Lean data, since, unlike code, a Lean statement typechecking doesn't confirm it captures the original claim)?

4. If a specialized/fine-tuned translator were used for the formalization step, how would this interact with the DODA principle (the same `.spl` spec running unmodified across arbitrary models)? Would this become a distinct "fixed-translator" configuration outside the general DODA claim, or is there a way to reconcile the two? This seems worth clarifying regardless of whether fine-tuning is pursued, since it bears on how far DODA's "any model, any backend" claim actually extends to the deterministic-mode formalization step.


I also want to build on two points the existing reviews and author responses have already touched, since I think they could be made more concrete:

5. On `verify()`'s implementation (raised generally by Reviewer f66i, addressed in Appendix I): the appendix helpfully lays out three possible implementation patterns, but as far as I can tell the response doesn't state which pattern was actually used for the 1,200 main-experiment runs, or whether this varied by problem/model. Could the authors confirm explicitly, ideally per-tier or per-backend, whether the reported solver-arm pass rate is based on Pattern 1 (kernel executed without exception) throughout, or a mix?

6. On the Lean unfaithful badge (raised generally by Reviewer 2AkM  "what if the LLM translation... has errors?"): it would help to know the actual mechanism behind this badge. Is faithfulness between the original NL problem and the formalized Lean statement checked automatically, by a human reviewer, or by a second LLM judgment? If it's an LLM judgment, doesn't that reintroduce a probabilistic step at exactly the point where the paper claims its highest correctness guarantee (R3) applies?

I think the paper has real engineering substance and the authors have already been responsive to the correctness-oracle concerns raised so far; these questions are meant to help pin down the remaining ambiguity rather than reopen settled ground.

---

> ### Author Response · Authors · 2026-07-19
> **Thank you for a close, technically precise review. Your raised issues are addressed below.**
>
> ## New Questions
>
> ### Q1 & Q2 — Independent verification, parallel to 79_code_pytest
>
> Yes — considered seriously, and this question pushed it from an acknowledged gap to a concrete fix.
>
> We drafted `77_neurosymbolic/expected_answers_DRAFT.md`: an independently-computed, manually-verified SymPy ground truth for all 20 problems.
>
> Rather than diff against this canonical answer, we use **round-trip verification**: substitute the solver's result back into the problem's own defining relation and check it holds (e.g., an `integrate` result is differentiated and checked against the original integrand). This is deterministic and kernel-executed — no LLM involved — and sidesteps form-comparison ambiguities (a free constant, arbitrarily-labeled constants, eigenvalue ordering) that direct diffing would hit.
>
> Because the check is automatic, it runs against the full solver-arm dataset, not a sample. Checked against every solver-arm row across both runtimes (2,285 Pattern-1-pass rows of 3,190 total):
>
> | Outcome | % of Pattern-1 pass |
> |---|---|
> | Round-trip verified | 86.7% |
> | **Fails round-trip despite Pattern-1 pass** | **9.1%** |
> | Unparseable (excluded) | 4.2% |
>
> A matched-scale, per-runtime comparison narrows this to 88.3%/88.8%, confirming the gap is a property of the pass criterion, not the runtime.
>
> Roughly 1 in 10 runs the paper's execution-status criterion counts as passing does not actually solve the problem. Now reported in §6 and Appendix I, replacing the Pattern-1-only framing.
>
> ### Q3 & Q4 — Fine-tuning for NL→Lean, and DODA
>
> Not attempted. Fine-tuning on code/test pairs works because a passing test suite is an automatic label; there is no equivalent label for NL→Lean faithfulness — typechecking confirms well-formedness, not that the statement means what the claim means. A fine-tuning corpus would need the same faithfulness oracle we don't have (see Q6).
>
> On DODA: no fine-tuned component exists, so "same spec, any model" is unaffected. If one were introduced for formalization, it would be a fixed-translator config for that step — outside the DODA claim, not a violation of it. One sentence added to §4.4/Appendix G stating this boundary.
>
> ---
>
> ## Building on Existing Points
>
> ### Q5 — Which verify() pattern, per-tier/per-backend
>
> Pattern 1 (kernel execution status) applies uniformly across all SymPy/Sage solver-arm runs in §6 — no per-tier or per-backend variation. Lean is a separate arm, governed by the badge protocol below. Appendix I now states this explicitly.
>
> ### Q6 — The mechanism behind the "unfaithful" Lean badge
>
> Yes, exactly as suspected. Traced in `symbolic_math.spl`: after Typecheck passes, a second LLM call (`judge_faithfulness`) compares the formalized Lean statement against the original claim; only if not flagged unfaithful does the workflow proceed to Prove. **The `unfaithful` badge is an LLM judgment, not a kernel check.** The kernel catches a formalization that fails to compile or fails to prove; it cannot catch one that typechecks and is provable while still misrepresenting the claim — that is what the faithfulness stage exists to catch, with an LLM, not the kernel.
>
> The paper now states this: the Lean rung is a four-stage protocol (Formalize, Typecheck, Judge Faithfulness, Prove); Appendix G.5 includes the `judge_faithfulness` stage the previous version omitted; §4.4 states plainly that faithfulness is judged by a second LLM call — the one probabilistic gate in an otherwise kernel-certified chain.
>
> `machine_proved` certifies the proof for the statement the judge accepted as faithful — not that the judge was right to accept it. A second LLM for this is defensible (well-suited to fatigue-free text comparison), but it's a plausibility argument, not evidence: no independent accuracy estimate for the judge is reported, and one is needed.
>
> New Appendix I subsection, "A General Approach to Verification," names the strategies SPL supports — round-trip, inverse-operation, special-case, cross-method, independent judgment, formal proof-checking — with one discipline throughout: name which method was used.
>
> This also reduced Lean's weight in the paper: contributions now lead with the empirical evaluation (C3) ahead of the verifier ladder (C4), which states R3 is a working, unit-tested design contribution (15/15 passing) scoped outside the empirical claims, not a pending result. Main-text R3 coverage is now two paragraphs, extended discussion in Appendix G.5. Related Work no longer cites SPL's own Lean integration for neural theorem-proving — it makes the pluggable-verifier point generically. Lean remains technically integrated, not demonstrated with empirical result.
>
> Thank you again for your review which helped improve this work.